# Dynamic Token Normalization Improves Vision Transformer

**Wenqi Shao** [1,2]**, Yixiao Ge** [2]**, Zhaoyang Zhang** [1]**, Xuyuan Xu** [3]**,**
**Xiaogang Wang** [1]**, Ying Shan** [2]**, Ping Luo** [4]

{weqish@link,zhaoyangzhang@link, xgwang@ee.}cuhk.edu.hk
{yixiaoge,evanxyxu,yingsshan}@tencent.com    pluo.lhi@gmail.com
[1] The Chinese University of Hong Kong    [2] ARC Lab, Tencent PCG
[3] AI Technology Center of Tencent Video    [4] The University of Hong Kong

## Abstract

Vision Transformer (ViT) and its variants (e.g., Swin, PVT) have achieved great success in various computer vision tasks, owing to their capability to learn long-range contextual information. Layer Normalization (LN) is an essential ingredient in these models. However, we found that the ordinary LN makes tokens at different positions similar in magnitude because it normalizes embeddings within each token. It is difficult for Transformers to capture inductive bias such as the positional context in an image with LN. We tackle this problem by proposing a new normalizer, termed Dynamic Token Normalization (DTN), where normalization is performed both within each token (intra-token) and across different tokens (inter-token). DTN has several merits. Firstly, it is built on a unified formulation and thus can represent various existing normalization methods. Secondly, DTN learns to normalize tokens in both intra-token and inter-token manners, enabling Transformers to capture both the global contextual information and the local positional context. Thirdly, by simply replacing LN layers, DTN can be readily plugged into various vision transformers, such as ViT, Swin, PVT, LeViT, T2T-ViT, BigBird and Reformer. Extensive experiments show that the transformer equipped with DTN consistently outperforms baseline model with minimal extra parameters and computational overhead. For example, DTN outperforms LN by 0.5% - 1.2% top-1 accuracy on ImageNet, by 1.2 - 1.4 box AP in object detection on COCO benchmark, by 2.3% - 3.9% mCE in robustness experiments on ImageNet-C, and by 0.5% - 0.8% accuracy in Long ListOps on Long-Range Arena. Codes will be made public at `https://github.com/wqshao126/DTN`.

## 1 Introduction

Vision Transformers (ViTs) have been employed in various tasks of computer vision, such as image classification (Dosovitskiy et al., 2020; Yuan et al., 2021), object detection (Wang et al., 2021b; Liu et al., 2021) and semantic segmentation (Strudel et al., 2021). Compared with the conventional Convolutional Neural Networks (CNNs), ViTs have the advantages in modeling long-range dependencies, as well as learning from multimodal data due to the representational capacity of the multi-head self-attention (MHSA) modules (Vaswani et al., 2017; Dosovitskiy et al., 2020). These appealing properties are desirable for vision systems, enabling ViTs to serve as a versatile backbone for various visual tasks.

However, despite their great successes, ViTs often have greater demands on large-scale data than CNNs, due to the lack of inductive bias in ViTs such as local context for image modeling (Dosovitskiy et al., 2020). In contrast, the inductive bias can be easily induced in CNNs by sharing convolution kernels across pixels in images (Krizhevsky et al., 2012). Many recent advanced works attempt to train a data-efficient ViT by introducing convolutional prior. For example, by distilling knowledge from a convolution teacher such as RegNetY-16GF (Radosavovic et al., 2020), DeiT (Touvron et al., 2021) can attain competitive performance when it is trained on ImageNet only. Another line of works alleviates this problem by modifying the architecture of ViTs. For example, Swin (Liu

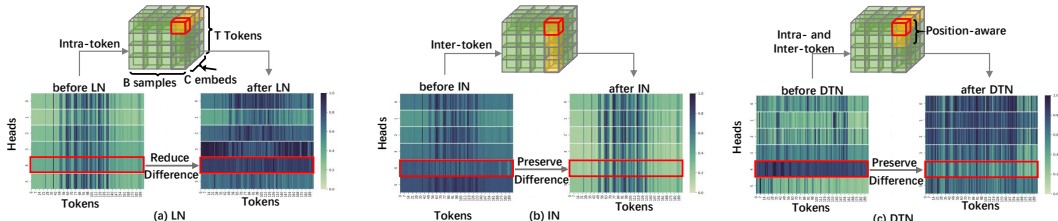

Figure 1: The visualization of token difference in magnitude when different normalizers are employed including LN (**a**), IN (**b**) and the proposed DTN (**c**). The results are obtained using a trained ViT-S with different normalizers on a randomly chose sample. The cube represent a feature of tokens whose dimension is $B \times T \times C$, and each token is a vector with $C$-dimension embedding. We express IN, LN and DTN by coloring different dimensions of those cubes. We use a heatmap to visualize the magnitude of all the tokens, i.e., the norm of token embedding for each head. (a) shows that LN operates within each token. Hence, it makes token magnitude have uniform magnitude regardless of their positions. Instead, (b) and (c) show that IN and our DTN can aggregate statistics across different tokens, thus preserving variation between different tokens.

et al., 2021) and PVT (Wang et al., 2021b) utilize a hierarchical representation to allow for hidden features with different resolutions, preserving the details of locality information. Although these works present excellent results in the low-data regime, they either require convolution or require tedious architectural design.

In this work, we investigate the problem of inductive bias in vision transformers from the perspective of the normalization method. Specifically, it is known that layer normalization (LN) (Ba et al., 2016) dominates various vision transformers. However, LN normalizes the embedding within each token, making all the tokens have similar magnitude regardless of their spatial positions as shown in Fig.1(a). Although LN encourages transformers to model global contextual information (Dosovitskiy et al., 2020), we find that transformers with LN cannot effectively capture the local context in an image as indicated in Fig.2(a), because the semantic difference between different tokens has been reduced.

To tackle the above issue, we propose a new normalizer for vision transformers, termed dynamic token normalization (DTN). Motivated by Fig.1(b), where normalization in an inter-token manner like instance normalization (IN) (Ulyanov et al., 2016) can preserve the variation between tokens, DTN calculates its statistics across different tokens. However, directly aggregating tokens at different positions may lead to inaccurate estimates of normalization constants due to the domain difference between tokens as shown in Fig.3. To avoid this problem, DTN not only collects intra-token statistics like LN, but also employs a position-aware weight matrix to aggregate tokens with similar semantic information, as illustrated in Fig.1(c). DTN has several attractive benefits. (1) DTN is built on a unified formulation, making it capable of representing various existing normalization methods such as LN and instance normalization (IN). (2) DTN learns to normalize embeddings in both intra-token and inter-token manners, thus encouraging transformers to capture both global contextual information and local positional context as shown in Fig.2(c). (3) DTN is fully compatible with various advanced vision transformers. For example, DTN can be easily plugged into recently proposed models such as PVT (Wang et al., 2021b) and Swin (Liu et al., 2021) by simply replacing LN layers in the original networks.

The main **contributions** of this work are three-fold. (1) From the perspective of the normalization, we observe that LN reduces the difference in magnitude between tokens regardless of their different spatial positions, making it ineffective for ViTs to induce inductive bias such as local context. (2) We develop a new normalization technique, namely DTN, for vision transformers to capture both long-range dependencies and local positional context. Our proposed DTN can be seamlessly plugged into various vision transformers, consistently outperforms its baseline models with various normalization methods such as LN. (3) Extensive experiment such as image classification on ImageNet (Russakovsky et al., 2015), robustness on ImageNet-C (Hendrycks & Dietterich, 2019), self-supervised pre-training on ViTs (Caron et al., 2021), ListOps on Long-Range Arena (Tay et al., 2021) show that DTN can achieve better performance with minimal extra parameters and marginal increase of computational overhead compared to existing approaches. For example, the variant of ViT-S with DTN exceeds its counterpart of LN by $1.1\%$ top-1 accuracy on ImageNet under the same amount of parameters with only $5.4\%$ increase of FLOPs.

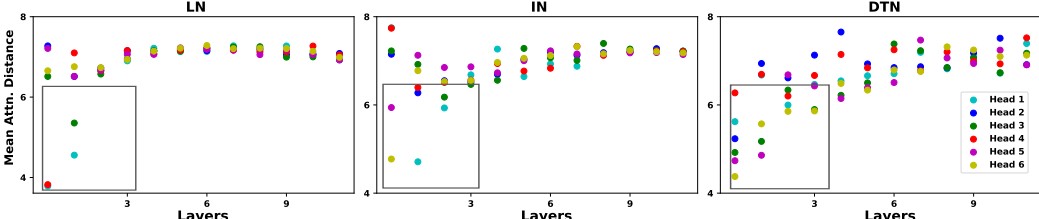

Figure 2: The visualization of mean attention distance in multi-head self-attention (MHSA) module of a trained ViT-S with (a) LN, (b) IN, and (c) DTN. The mean attention distance denotes the average number of patches between the center of attention, and the query patch Dosovitskiy et al. (2020) (see details in Appendix Sec. B.1). A large mean attention distance indicates that the head would attend to most image patches, presenting excellent ability of modeling long-range dependencies. (a) shows that LN can perform excellent long-range contextual modeling while failing to capture local positional context. (b & c) show that multiple heads in ViT-S with IN and DTN have a small mean attention distance. Hence, IN and our DTN can capture local context because they preserves the variation between tokens as shown in Fig.1(b & c).

## 2 RELATED WORK

**Vision transformers.** Vision transformer models have been widely utilized recently (Dosovitskiy et al., 2020; Yang et al., 2022). For example, the vision transformer (ViT) (Dosovitskiy et al., 2020) splits the source image into a sequence of patches and fed them into transformer encoders. Although ViT presents excellent results on image classification, it requires costly pre-training on large-scale datasets (e.g., JFT-300M (Sun et al., 2017)) as ViT models are hard to capture inductive bias. Many works tackle this problem by introducing convolutional priors into ViTs because convolution inherently encodes inductive bias such as local context. For example, DeiT Touvron et al. (2021) uses a token-based strategy to distill knowledge from convolutional teachers. Moreover, PVT (Wang et al., 2021b), Swin (Liu et al., 2021), T2T (Yuan et al., 2021), CVT (Wu et al., 2021) and et al. introduces hierarchical representation to vision transformers by modifying the structure of transformers. Different from these works, our DTN trains data-efficient transformers from the perspective of normalization. We show that DTN enables vision transformers to capture both global context and local positional locality. More importantly, DTN is a generally functional module and can be easily plugged into the aforementioned advanced vision transformers.

**Normalization methods.** Normalization techniques have been extensively investigated in CNN (Ioffe & Szegedy, 2015; Shao et al., 2019). For different vision tasks, various normalizers such as BN (Ioffe & Szegedy, 2015), IN (Ulyanov et al., 2016), LN (Ba et al., 2016), GN (Wu & He, 2018), SN (Luo et al., 2019a) and et al. are developed. For example, BN is widely used in CNN for image classification, while IN performs well in pixel-level tasks such as image style transfer. To our interest, LN outperforms the above normalizers in transformers and has dominated various transformer-based models. Although ScaleNorm (Nguyen & Salazar, 2019) and PowerNorm (Shen et al., 2020) improves LN in language tasks such as machine translation, it does not work well in vision transformer as shown in Table 3. Instead, we observe that vision transformer models with LN cannot effectively encode local context in an image, as shown in Fig.2. To resolve this issue, we propose a new normalizer named DTN, which can capture both global contextual information and local positional context as shown in Fig.2(c).

**Dynamic Architectures.** Our work is related to dynamic architectures such as mixture-of-experts (MoE) (Shazeer et al., 2018; Eigen et al., 2013) and dynamic channel gating and grouping networks (Hua et al., 2018; Zhang et al., 2019). Similar to these works, DTN also learns weight ratios to select computation units. In DTN, such a strategy is very effective to combine intra- and inter-token statistics. In addition, DTN is substantially different from DN (Luo et al., 2019b), where dynamic normalization can be constructed as the statistics in BN and LN can be inferred from the statistics of IN. However, it does not hold in the transformer because LN normalizes within each token embedding. Compared to SN (Luo et al., 2019a), we get rid of BN, which is empirically detrimental to ViTs. Moreover, we use a position-aware probability matrix to collect intra-token statistics, making our DTN rich enough to contain various normalization methods.

## 3 METHOD

This section firstly revisits layer normalization (LN) (Ba et al., 2016) and instance normalization (IN) (Ulyanov et al., 2016) and then introduces the proposed dynamic token normalization (DTN).

### 3.1 REVISITING LAYER NORMALIZATION

In general, we denote a scalar with a regular letter (e.g. 'a') and a tensor with a bold letter (e.g. $\boldsymbol{a}$). Let us begin by introducing LN in the context of ViTs. We consider a typical feature of tokens $\boldsymbol{x} \in \mathbb{R}^{B \times T \times C}$ in the transformer where $B, T$, and $C$ denote batch size, token length, embedding dimension of each token respectively, as shown in Fig.1(a). Since LN is performed in a data-independent manner, we drop the notation of '$B$' for simplicity.

LN standardizes the input feature by removing each token's mean and standard deviation and then utilizes an affine transformation to obtain the output tokens. The formulation of LN is written by

$$\tilde{x}_{tc} = \gamma_c \frac{x_{tc} - \mu_t^{ln}}{\sqrt{(\sigma^2)^{ln} + \epsilon}} + \beta_c \tag{1}$$

where $t$ and $c$ are indices of tokens and embeddings of a token respectively, $\epsilon$ is a small positive constant to avoid zero denominator, and $\gamma_c, \beta_c$ are two learnable parameters in affine transformation. In Eqn.(1), the normalization constants of LN $\mu_t^{ln}$ and $(\sigma^2)_t^{ln}$ are calculated in an intra-token manner as shown in Fig.1(a). Hence, for all $t \in [T]$, we have

$$\mu_t^{ln} = \frac{1}{C} \sum_{c=1}^{C} x_{tc} \quad \text{and} \quad (\sigma^2)_t^{ln} = \frac{1}{C} \sum_{c=1}^{C} (x_{tc} - \mu_t^{ln})^2 \tag{2}$$

Previous works show that LN works particularly well with the multi-head self-attention (MHSA) module to capture long-range dependencies in vision tasks, as can also be seen from Fig.2(a) where most heads in MHSA after LN attend to most of the image. However, we find that LN reduces the difference in magnitude between tokens at different positions, preventing the MHSA module from inducing inductive bias such as local context. To see this, Eqn.(2) shows that the mean and variance are obtained within each token, implying that each token would have zero mean and unit variance. Further, the standardized tokens are then operated by the same set of affine parameters $\{\gamma_c, \beta_c\}_{c=1}^{C}$ through Eqn(1). Therefore, all the tokens returned by LN would have a similar magnitude regardless of their positions in the image.

This fact can also be observed by visualizing token magnitude in Fig.1(a). As we can see, the difference between tokens after LN is reduced. However, since tokens are generated from image patches at different spatial locations, they should encode specific semantic information to embed the local context in an image. As a result, MHSA module after LN cannot effectively capture local context as presented in Fig.2(a) where only a few heads have a small attention distance.

Recent works tackle the issue of inductive bias in ViTs by combining the design of convolutional neural network (CNN) and vision transformers (Wang et al., 2021b; Liu et al., 2021). Although these methods have achieved good performance on various vision tasks, they still require complicated architecture design. In this paper, we aim to improve vision transformers by designing a new normalization method.

**Instance Normalization (IN).** IN provides an alternative to normalizing tokens while preserving the variation between them. In CNN, IN learns invariant features to a pixel-level perturbation, such as color and styles. Hence, IN can be employed to learn features of tokens with the local context. The definition of IN is the same with LN in Eqn.(1) except for the acquisition of normalization constants,

$$\mu_c^{in} = \frac{1}{T} \sum_{t=1}^{T} x_{tc} \quad (\sigma^2)_c^{in} = \frac{1}{T} \sum_{t=1}^{T} (x_{tc} - \mu_c^{in})^2. \tag{3}$$

Since IN obtains its statistics in an inter-token manner, the tokens returned by IN still preserves the variation between tokens in each head as shown in Fig.1(b). In addition, As we can observe from Fig.2(b), MHSA in the transformer with IN have more heads with a small mean attention distance than that of LN, showing that IN encourages MHSA to model local context.

However, IN has a major drawback in the context of vision transformers. From Fig.3(a), the normalization constants in IN are acquired by considering all tokens in the image. However, tokens

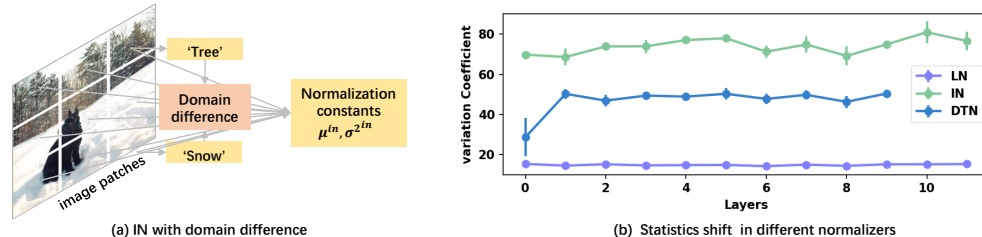

(a) IN with domain difference          (b) Statistics shift in different normalizers

Figure 3: **(a)** illustrates the normalization constants in IN. IN obtains its normalization constants by considering all patches from the image. Hence the semantic distribution shift from different image patches may lead to inaccurate estimates of normalization constants in IN. **(b)** shows the relative variation between token features and the statistics (mean), which is measured by the variation coefficient defined by the norm of $|(\boldsymbol{x} - \boldsymbol{\mu})/\boldsymbol{x}|$ where $\boldsymbol{\mu}$ is the mean in normalization layer. A larger variation coefficient indicates that the mean is further away from the token feature. From (b), IN may lead to inaccurate estimates due to the domain difference between tokens. LN and our proposed DTN can mitigate the problem of inaccurate estimates of statistics in IN.

encoded from different image patches in the vision transformer may come from different semantic domains. For example, the top patch presents a tree, while the bottom is a snow picture. If we take an average over all the tokens to calculate mean and variance, it would lead to inaccurate estimates of normalization constants as shown in Fig.3(b). Empirically, we find that ViT-S with IN obtains much lower top-1 accuracy (77.7%) on ImageNet than its counterpart with LN (79.9%). Section 3.2 resolves these challenges by presenting dynamic token normalization.

## 3.2 Dynamic Token Normalization (DTN)

DTN encourages attention heads in MHSA to model both global and local context by inheriting the advantages of LN and IN.

**Definition.** DTN is defined by a unified formulation. Given the feature of tokens $\boldsymbol{x} \in \mathbb{R}^{T \times C}$, DTN normalizes it through

$$\tilde{\boldsymbol{x}} = \boldsymbol{\gamma} \frac{\boldsymbol{x} - \mathrm{Concate}_{h \in [H]}\{\boldsymbol{\mu}^h\}}{\sqrt{\mathrm{Concate}_{h \in [H]}\{(\boldsymbol{\sigma}^2)^h\} + \epsilon}} + \boldsymbol{\beta} \tag{4}$$

where $\boldsymbol{\gamma}$, $\boldsymbol{\beta}$ are two C-by-1 vectors by stacking all $\gamma_c$ and $\beta_c$ into a column, and $\boldsymbol{\mu}^h \in \mathbb{R}^{T \times \frac{C}{H}}$, $(\boldsymbol{\sigma}^2)^h \in \mathbb{R}^{T \times \frac{C}{H}}$ are normalization constants of DTN in head $h$ where $H$ denotes the number of heads in transformer. The 'Concate' notation indicates that DTN concatenates normalization constants from different heads. This design is motivated by two observations in Fig.2. First, attention heads attend to patches in the image with different attention distances, encouraging diverse contextual modeling for different self-attention heads. Second, by obtaining statistics in two different ways (intra-token and inter-token), LN and IN produce different patterns of attention distance in MHSA. Hence, we design DTN by calculating normalization constants specific to each head.

**Normalization constants in DTN.** As aforementioned in Sec.3.1, LN acquires its normalization constants within each token, which is helpful for global contextual modeling but fails to capture local context between tokens. Although IN can achieve self-attention with locality, it calculates the normalization constants across tokens, resulting in inaccurate mean and variance estimates. To overcome the above difficulties, DTN obtains normalization constants by trading off intra- and inter-token statistics as given by

$$\begin{aligned} \boldsymbol{\mu}^h &= \lambda^h (\boldsymbol{\mu}^{ln})^h + (1 - \lambda^h) \boldsymbol{P}^h \boldsymbol{x}^h, \\ (\boldsymbol{\sigma}^2)^h &= \lambda^h ((\boldsymbol{\sigma}^2)^{ln})^h + (1 - \lambda^h)[\boldsymbol{P}^h(\boldsymbol{x}^h \odot \boldsymbol{x}^h) - (\boldsymbol{P}^h \boldsymbol{x}^h \odot \boldsymbol{P}^h \boldsymbol{x}^h)] \end{aligned} \tag{5}$$

where $(\boldsymbol{\mu}^{ln})^h \in \mathbb{R}^{T \times \frac{C}{H}}$, $((\boldsymbol{\sigma}^2)^{ln})^h \in \mathbb{R}^{T \times \frac{C}{H}}$ are intra-token mean and variance obtained by stacking all $\mu_t^{ln}$ in Eqn.(2) into a column and then broadcasting it for $C/H$ columns, $\boldsymbol{x}^h \in \mathbb{R}^{T \times \frac{C}{H}}$ represents token embeddings in the head $h$ of $\boldsymbol{x}$. In Eqn.(5), $\boldsymbol{P}^h \boldsymbol{x}^h \in \mathbb{R}^{T \times \frac{C}{H}}$, $[\boldsymbol{P}^h(\boldsymbol{x}^h \odot \boldsymbol{x}^h) - (\boldsymbol{P}^h \boldsymbol{x}^h \odot \boldsymbol{P}^h \boldsymbol{x}^h)]$ are expected to represent inter-token mean and variance respectively. Towards this goal, we define $\boldsymbol{P}^h$ as a T-by-T learnable matrix satisfying that the sum of each row equals 1. For example, when $\boldsymbol{P}^h = \frac{1}{T}\mathbf{1}$ and $\mathbf{1}$ is a T-by-T matrix with all ones, they become mean and variance of IN respectively. Moreover, DTN utilizes a learnable weight ratio $\lambda^h \in [0, 1]$ to trade off intra-token and inter-token

statistics. By combining intra-token and inter-token normalization constants for each head, DTN not only preserves the difference between different tokens as shown in Fig.1(c), but also enables different attention heads to perform diverse contextual modelling in MHSA as shown in Fig.2(c). In DTN, the weight ratios for $\boldsymbol{\mu}^h$ and $\boldsymbol{\sigma}^h$ can be different as the mean and variance in the normalization plays different roles in network's training Luo et al. (2018); Xu et al. (2019), but they are shared in Eqn.(5) to simplify the notations.

**Representation Capacity.** By comparing Eqn.(5) and Eqn.(2-3), we see that DTN calculates its normalization constant in a unified formulation, making it capable to represent a series of normalization methods. For example, when $\lambda^h = 1$, we have $\boldsymbol{\mu}^h = (\boldsymbol{\mu}^{ln})^h$ and $(\boldsymbol{\sigma}^2)^h = ((\boldsymbol{\sigma}^2)^{ln})^h$. Hence, DTN degrades into LN. When $\lambda^h = 0$ and $\boldsymbol{P}^h = \frac{1}{T}\mathbf{1}$, we have $\boldsymbol{\mu}^h = \frac{1}{T}\mathbf{1}\boldsymbol{x}^h$ and $(\boldsymbol{\sigma}^2)^h = \frac{1}{T}\mathbf{1}(\boldsymbol{x}^h \odot \boldsymbol{x}^h) - (\frac{1}{T}\mathbf{1}\boldsymbol{x}^h \odot \frac{1}{T}\mathbf{1}\boldsymbol{x}^h)$ which are the matrix forms of mean and variance in Eqn.(3). Therefore, DTN becomes IN in this case. Moreover, when $\lambda^h = 0$ and $\boldsymbol{P}^h$ is a normalized $k$-banded matrix with entries in the diagonal band of all ones, DTN becomes a local version of IN which obtains the mean and variance for each token by only looking at $k$ neighbouring tokens. Some examples of $k$-banded matrix is shown in Fig.4 of Appendix.

The flexibility of $\lambda^h$ and $\boldsymbol{P}^h$ enables DTN to mitigate the problem of inaccurate estimates of statistics in IN in two ways. Firstly, by giving weight ratio $\lambda$ a large value, DTN behaves more like LN, which does not suffer from inaccurate estimates of normalization constants. Secondly, by restricting the entries of $\boldsymbol{P}^h$ such as $k$-banded matrix, DTN can calculate the statistics across the tokens that share similar semantic information with the underlying token. As shown in Fig.3(b), the relative variation between tokens and the corresponding statistics (mean) in DTN is significantly reduced compared with that of IN.

However, it is challenging to directly train a huge matrix with a special structure such as $k$-banded matrix through the vanilla SGD or Adam algorithm. Moreover, the extra learnable parameters introduced by $\boldsymbol{P}^h$ in DTN are non-negligible. Specifically, every $\boldsymbol{P}^h$ has $T^2$ variables, and there are often hundreds of heads in a transformer. Therefore, directly training $\boldsymbol{P}^h$ would introduce millions of parameters. Instead, we design $\boldsymbol{P}^h$ by exploring the prior of the relative position of image patches which leads to marginal extra parameters.

**Construction of $\boldsymbol{P}^h$.** We exploit the implicit visual clue in an image to construct $\boldsymbol{P}^h$. As shown in Fig.3(a), the closer two image patches are to each other, the more similar their semantic information would be. Therefore, we employ positional self-attention with relative positional embedding (d'Ascoli et al., 2021) to generate positional attention matrix $\boldsymbol{P}^h$ as given by

$$\boldsymbol{P}^h = \text{softmax}(\boldsymbol{R}\boldsymbol{a}^h) \tag{6}$$

where $\boldsymbol{R} \in \mathbb{R}^{T \times T \times 3}$ is a constant tensor representing the relative positional embedding. To embed the relative position between image patches, we instantiate $\boldsymbol{R}_{ij}$ as written by $\boldsymbol{R}_{ij} = [(\delta_{ij}^x)^2 + (\delta_{ij}^y)^2, \delta_{ij}^x, \delta_{ij}^y]^\mathsf{T}$ where $\delta_{ij}^x$ and $\delta_{ij}^y$ are relative horizontal and vertical shifts between patch $i$ and patch $j$ (Cordonnier et al., 2019), respectively. An example of the construction of $\boldsymbol{R}$ is illustrated in Fig.5 of Appendix. Moreover, $\boldsymbol{a}^h \in \mathbb{R}^{3 \times 1}$ are learnable parameters for each head. Hence, Eqn.(6) only introduces 3 parameters for each head, which is negligible compared to the parameters of the transformer model. In particular, by initializing $\boldsymbol{a}^h$ as equation below, $\boldsymbol{P}^h$ gives larger weights to tokens in the neighbourhood of size $\sqrt{H} \times \sqrt{H}$ relative to the underlying token,

$$\boldsymbol{a}^h = [-1, 2\Delta_1^h, 2\Delta_2^h]^\mathsf{T} \tag{7}$$

where $\boldsymbol{\Delta}^h = [\Delta_1^h, \Delta_2^h]$ is each of the possible positional offsets of the neighbourhood of size $\sqrt{H} \times \sqrt{H}$ relative to the underlying token. It also denotes the center of attention matrix $\boldsymbol{P}^h$, which means that $\boldsymbol{P}^h$ would assign the largest weight to the position $\boldsymbol{\Delta}^h$. For example, there are 4 optional attention centres (denoted by red box) when $H = 4$ as shown in Fig.6. Since the weights of each $\boldsymbol{P}^h$ concentrates on a neighbourhood of size $\sqrt{H} \times \sqrt{H}$ Eqn.(6-7), DTN can aggregates the statistics across tokens with similar semantic information through positional attention matrix $\boldsymbol{P}^h$.

## 3.3 DISCUSSION

**Implementation of DTN.** DTN can be inserted extensively into various vision transformer models by replacing LN in the original network. In this work, we verify the effectiveness of DTN on ViT, PVT, and Swin. However, it can also be easily applied to recently-proposed advanced vision

Table 1: Performance of the proposed DTN evaluated on ViT models with different sizes. DTN can consistently outperform LN with various ViT models. $H$ and $C$ denote the number of heads and the dimension of embeddings of each token, respectively.

| Model | Method | $H$ | $C$ | FLOPs | Params | Top-1 (%) | Top-5 (%) |
|---|---|---|---|---|---|---|---|
| ViT-T | LN | 3 | 192 | 1.26G | 5.7M | 72.2 | 91.3 |
| ViT-T* | LN | 4 | 192 | 1.26G | 5.7M | 72.3 | 91.4 |
| | DTN | 4 | 192 | 1.40G | 5.7M | **73.2** | **91.7** |
| ViT-S | LN | 6 | 384 | 4.60G | 22.1M | 79.9 | 95.0 |
| | DTN | 6 | 384 | 4.88G | 22.1M | **80.6** | **95.3** |
| ViT-S* | LN | 9 | 432 | 5.77G | 27.8M | 80.6 | 95.2 |
| | DTN | 9 | 432 | 6.08G | 27.8M | **81.7** | **95.8** |
| ViT-B | LN | 12 | 768 | 17.58G | 86.5M | 81.8 | 95.9 |
| | DTN | 12 | 768 | 18.13G | 86.5M | **82.3** | **96.0** |
| ViT-B* | LN | 16 | 768 | 17.58G | 86.5M | 81.7 | 95.8 |
| | DTN | 16 | 768 | 18.13G | 86.5M | **82.5** | **96.1** |

Table 2: Performance of the proposed DTN on various vision transformers in terms of accuracy and FLOPs on ImageNet. Our DTN improves the top-1 and top-5 accuracies by a clear margin compared with baseline using LN with a marginal increase of FLOPs.

| | PVT-Tiny | | | PVT-Small | | | Swin-T | | |
|---|---|---|---|---|---|---|---|---|---|
| | Top-1 | Top-5 | FLOPs | Top-1 | Top-5 | FLOPs | Top-1 | Top-5 | FLOPs |
| Baseline | 75.1 | 92.3 | 1.90G | 79.9 | 95.0 | 3.80G | 81.2 | 95.5 | 4.51G |
| DTN | **76.3** | **93.1** | 2.05G | **80.8** | **95.6** | 4.15G | **81.9** | **96.0** | 5.09G |

Table 3: Comparisons between DTN and other normalization methods, including LN, BN, IN, GN, ScaleNorm, and PowerNorm on ImageNet in terms of top-1 accuracy. Our DTN achieves competitive results over various normalizers.

| | LN | BN | IN | GN | SN | ScaleNorm | PowerNorm | DTN |
|---|---|---|---|---|---|---|---|---|
| ViT-S | 79.9 | 77.3 | 77.7 | 78.3 | 80.1 | 80.0 | 79.8 | **80.6** |
| $\Delta$ versus LN | - | -2.6 | -2.2 | -1.6 | +0.2 | +0.1 | -0.1 | **+0.7** |
| ViT-S* | 80.6 | 77.2 | 77.6 | 79.5 | 81.0 | 80.6 | 80.4 | **81.7** |
| $\Delta$ versus LN | - | -3.4 | -3.0 | -1.1 | +0.4 | 0.0 | -0.2 | **+1.1** |

transformers such as CVT (Wu et al., 2021). As shown in Algorithm 1 of Appendix, DTN can be implemented in a forward pass. The introduced learnable parameters by DTN are weight ratio $\lambda^h$ and $\boldsymbol{a}^h$ by Eqn.(7) which are initialized through line 2 of Algorithm 1. Note that All the computations involving in DTN are differentiable.

**Complexity analysis.** The computation and parameter complexity for different normalization methods are compared in Table 7b. Note that the number of heads in the transformer is far less than the embedding dimension i.e., $H \ll C$. DTN introduces almost the same learnable parameters compared with IN and LN. Moreover, our DTN calculates inter-token normalization constants for each token embedding through positional attention matrix $\boldsymbol{P}^h$ in Eqn.(5), resulting in $O(BCT^2)$ computational complexity. The computational overhead proportional to $T^2$ is nonnegligible when the token length $T$ is large. To make the computation efficient, we adopt a global average pooling operation with pooling size $s$ on the token level, which reduces the length of tokens to $T/s^2$. Hence, the computational complexity of DTN decreases to $O(BCT^2/s^4)$. We also investigate the FLOPs of vision transformers with DTN in Table 1 and Table 2.

## 4 EXPERIMENT

This section extensively evaluates the effectiveness of the proposed Dynamic Token Normalization (DTN) in various computer vision tasks, including classification on ImageNet, robustness on ImageNet-C and ImageNet-R (Sec.B.2 of Appendix), and self-supervised pre-training (Sec.B.3 of Appendix). Besides, we also test DTN on the task of ListOps using transformers with different efficient self-attention modules. An ablation study is provided to analyze our proposed DTN comprehensively (see more details in Sec.B.4 of Appendix). The training details can be found in Sec.B.1.

Table 4: Comparisons between DTN and other normalization methods on Long ListOps task of LRA in terms of accuracy. Our DTN is validated on three SOTA transformers architectures and yields competitive results to other normalizations.

|  | LN | BN | IN | GN | ScaleNorm | DTN |
|---|---|---|---|---|---|---|
| Transformer (Vaswani et al., 2017) | 36.4 | 29.2 | 22.7 | 28.2 | 36.6 | **37.0** |
| Δ versus LN | - | -7.2 | -13.7 | -8.2 | +0.2 | **+0.6** |
| BigBird (Zaheer et al., 2020) | 36.7 | 36.7 | 36.3 | 36.8 | 36.9 | **37.5** |
| Δ versus LN | - | 0.0 | -0.4 | +0.1 | +0.2 | **+0.8** |
| Reformer (Kitaev et al., 2020) | 37.3 | 37.2 | 37.0 | 37.4 | 37.3 | **37.8** |
| Δ versus LN | - | -0.1 | -0.3 | +0.1 | 0.0 | **+0.5** |

## 4.1 RESULTS ON IMAGENET

**DTN on ViT models with different sizes.** We evaluate the performance of the proposed DTN on ViT models with different sizes, including ViT-T, ViT-S, and ViT-B. Note that DTN obtains inter-token normalization constants by positional attention matrix $P^h$ in Eqn.(5), which gives larger weights to entries in the neighbourhood of size $\sqrt{H} \times \sqrt{H}$ where $H$ denotes the number of heads. To exploit this property, we increase the number of heads from $6$ to $9$ in order to produce $3 \times 3$ neighborhood and decrease the embedding dimension in each head from $64$ to $48$. The resulting models are denoted by ViT-T$^*$, ViT-S$^*$, and ViT-B$^*$.

As shown in Table 1, ViT models with DTN can consistently outperform the baseline models using LN with a marginal increase of parameters and FLOPs on the ImageNet dataset, demonstrating the effectiveness of our DTN. For example, DTN surpasses LN by a margin of $1.1\%$ top-1 accuracy on ViT-S$^*$ with almost the same parameters and only $5.4\%$ increase of FLOPs compared with LN. On the other hand, we see the margin of performance becomes larger when ViT models are instantiated with a larger number of heads. For instance, DTN exceeds LN by $0.8\%$ top-1 accuracy on ViT-B$^*$ while DTN outperforms LN by $0.5\%$ top-1 accuracy on ViT-B.

**DTN on various vision transformers.** We investigate the effect of DTN on various vision transformers on ImageNet. Since normalization is an indispensable and lightweight component in transformers, it is flexible to plug the proposed DTN into various vision transformers by directly replacing LN. In practice, two representative vision transformers, including PVT Wang et al. (2021b) and Swin Liu et al. (2021) are selected to verify the versatility of our DTN. We use their respective open-sourced training framework. As shown in Table 2, DTN improves the performance of PVT and Swin by a clear margin in terms of top-1 and top-5 accuracies compared with baseline using LN with a marginal increase of FLOPs. Specifically, PVT-Tiny, PVT-Small, and Swin-T with DTN surpass their baseline with LN by $1.2\%$, $0.9\%$, and $0.7\%$ top-1 accuracy, respectively.

**DTN versus other normalization methods.** We show the advantage of our DTN over exiting representative normalizers that are widely utilized in computer vision models such as BN, IN, and GN, and in language models such as LN, ScaleNorm, and PowerNorm. The results are reported in Table 3. It can be seen that our DTN achieves competitive performance compared with other normalization methods on both ViT-S and ViT-S$^*$. In particular, we see that vanilla BN and IN obtain the worst performance, which is inferior to LN by $2.6\%$ and $2.2\%$ top-1 accuracy, respectively. This may be caused by inaccurate estimates of mean and variance in IN and BN. For example, PowerNorm improves BN by providing accurate normalization constant through moving average in both forward and backward propagation, achieving comparable top-1 accuracy compared with LN. Our proposed DTN further calculates the statistics in both intra-token and inter-token ways, consistently outperforming previous normalizers.

## 4.2 RESULTS ON LISTOPS OF LONG-RANGE ARENA

We further evaluate our DTN on the ListOps task of Long-Range Arena benchmark (Tay et al., 2021) to validate the proposed methods on non-image sources. The Long-Range Arena (LRA) benchmark is designed explicitly to assess efficient Transformer models under the long-context scenario. To demonstrate the effectiveness of DTN, we choose the Long ListOps task, one of the most challenging tasks in the LRA benchmark. The Long Listops task is a longer variation of standard ListOps task (Nangia & Bowman, 2018) with up to 2K sequence lengths, which is considerably difficult. We compare DTN with other normalizations using vanilla Transformer (Vaswani et al., 2017) and

two recently proposed efficient Transformer models including BigBird (Zaheer et al., 2020) and Reformer (Kitaev et al., 2020). All of the experiments are conducted with default settings in (Tay et al., 2021) and the results are reported in Table. 4. As shown, DTN yields consistent performance gains ($> 0.5\%$) compared to other listed normalizations, including the SOTA method Reformer. For the other normalizations, we observe a significant accuracy drop on vanilla Transformer models with IN, BN, and GN, which is possibly caused inaccurate estimates of statistics (as mentioned in Sec.3.1).

### 4.3 ABLATION STUDY

**Effect of each component.** As shown in Eqn.(5), DTN obtains its normalization constants by three crucial components including inter-token statistics from LN, intra-token statistics controlled by positional attention matrix $\boldsymbol{P}^h$, and learnable weight ratio $\lambda^h$. We investigate the effect of each compo-

Table 5: The effect of each component in DTN. Each component is crucial to the effectiveness of DTN.

|  | (a) LN | (b) IN | (c) | (d) | (e) DTN |
|---|---|---|---|---|---|
| $\lambda^h$ | 1.0 | 0.0 | 0.0 | 0.5 | learnable |
| $\boldsymbol{P}^h$ | - | $\frac{1}{T}\mathbf{1}$ | learnable | learnable | learnable |
| Top-1 | 80.6 | 77.6 | 81.2 | 81.3 | **81.7** |

nents by considering five variants of DTN. (a) $\lambda^h = 1$. In this case, DTN becomes LN; (b) $\lambda^h = 0$ and $\boldsymbol{P}^h = \frac{1}{T}\mathbf{1}$. For this situation, DTN turns into IN; (c) $\lambda^h = 0$ and $\boldsymbol{P}^h$ is learnable. (d) $\lambda^h = 0.5$ and $\boldsymbol{P}^h$ is learnable; (e) Both $\lambda^h$ and $\boldsymbol{P}^h$ are learnable which is our proposed DTN. The results are reported in Table 5. By comparing (b) and (a & c), we find that both intra-token statistics from LN and positional attention matrix $\boldsymbol{P}^h$ can improve IN by a large margin, showing the reasonableness of our design in Eqn.(5). It also implies that both intra-token and inter-token statistics are useful in normalization. By comparing (e) and (c & d), we see that a learnable $\lambda^h$ can better trade off inter-token and intra-token statistics for each head, which further improves the performance of DTN.

**Different initilization of $\boldsymbol{a}^h$.** We investigate the effect of different initializations of $\boldsymbol{a}^h$ using ViT-T* with DTN. To compare with the initialization in Eqn.(7), we also use a truncated normal distribution to initialize $\boldsymbol{a}^h$. In experiment, we find DTN with the initialization of $\boldsymbol{a}^h$ followed truncated normal distribution achieves 72.7% top-1 accuracy, which is worse than DTN with $\boldsymbol{a}^h$ initialized by Eqn.(7) (73.2% top-1 accuracy). The effectiveness of initialization in Eqn.(7) is further revealed by visualization in Fig.6. We can see that $\boldsymbol{P}^h$ generated by Eqn.(7) gives larger weights to its neighbouring tokens before training while $\boldsymbol{P}^h$ obtained by initialization with truncated normal distribution has uniform weights, indicating that Eqn.(7) helps DTN aggregate statistics over tokens with similar local context. After training, the initialization in Eqn.(7) can also better capture local positional content than initialization using truncated normal as shown in Fig.7.

## 5 CONCLUSION

In this work, we find that layer normalization (LN) makes tokens similar to each other regardless of their positions in spatial. It would result in the lack of inductive bias such as local context for ViTs. We tackle this problem by proposing a new normalizer named dynamic token normalization (DTN) where normalization constants are aggregated on intra- and inter-token bases. DTN provides a holistic formulation by representing a series of existing normalizers such as IN and LN. Since DTN considers tokens from different positions, it preserves the variation between tokens and thus can capture the local context in the image. Through extensive experiments and studies, DTN can adapt to ViTs with different sizes, various vision transformers, and tasks outperforming their counterparts. In particular, DTN improves the modeling capability of the self-attention module by designing a new normalizer, shedding light on future work on transformer-based architecture development. For example, DTN could be combined with a sparse self-attention module (Tang et al., 2022) because it encourages self-attention with a small attention distance, as shown in Fig.2.

**Acknowledgments.** We thank anonymous reviewers from the venue where we have submitted this work for their instructive feedback. This work is supported in part by Centre for Perceptual and Interactive Intelligence Limited, in part by the General Research Fund through the Research Grants Council of Hong Kong under Grants (Nos. 14203118, 14208619), in part by Research Impact Fund Grant No. R5001-18. Ping Luo is supported by the General Research Fund of HK No.27208720 and 17212120.

**Ethics Statement.** We improve the performance of vision transformers by the proposed dynamic token normalization (DTN). We notice that DTN can be plugged into various deep architectures and applied in a wide range of learning tasks. Hence, our work and AI applications in different tasks would have the same negative impact on ethics. Moreover, DTN may have different effects on different image sources, thus potentially producing unfair models. We will carefully study our DTN on the fairness of the model output.

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

$$\begin{bmatrix} 1 & 1 & 1 & 1 & 1 & 1 & 1 & 1 \\ 1 & 1 & 1 & 1 & 1 & 1 & 1 & 1 \\ 1 & 1 & 1 & 1 & 1 & 1 & 1 & 1 \\ 1 & 1 & 1 & 1 & 1 & 1 & 1 & 1 \\ 1 & 1 & 1 & 1 & 1 & 1 & 1 & 1 \\ 1 & 1 & 1 & 1 & 1 & 1 & 1 & 1 \\ 1 & 1 & 1 & 1 & 1 & 1 & 1 & 1 \\ 1 & 1 & 1 & 1 & 1 & 1 & 1 & 1 \end{bmatrix} \quad \begin{bmatrix} 1 & 1 & 0 & 0 & 0 & 0 & 0 & 0 \\ 1 & 1 & 1 & 0 & 0 & 0 & 0 & 0 \\ 0 & 1 & 1 & 1 & 0 & 0 & 0 & 0 \\ 0 & 0 & 1 & 1 & 1 & 0 & 0 & 0 \\ 0 & 0 & 0 & 1 & 1 & 1 & 0 & 0 \\ 0 & 0 & 0 & 0 & 1 & 1 & 1 & 0 \\ 0 & 0 & 0 & 0 & 0 & 1 & 1 & 1 \\ 0 & 0 & 0 & 0 & 0 & 0 & 1 & 1 \end{bmatrix} \quad \begin{bmatrix} 1 & 1 & 1 & 0 & 0 & 0 & 0 & 0 \\ 1 & 1 & 1 & 1 & 0 & 0 & 0 & 0 \\ 1 & 1 & 1 & 1 & 1 & 0 & 0 & 0 \\ 0 & 1 & 1 & 1 & 1 & 1 & 0 & 0 \\ 0 & 0 & 1 & 1 & 1 & 1 & 1 & 0 \\ 0 & 0 & 0 & 1 & 1 & 1 & 1 & 1 \\ 0 & 0 & 0 & 0 & 1 & 1 & 1 & 1 \\ 0 & 0 & 0 & 0 & 0 & 1 & 1 & 1 \end{bmatrix}$$

(a) Full-banded matrix with all ones     (b) 3-banded matrix with binary entries     (c) 5-banded matrix with binary entries

Figure 4: Illustration of different types of the banded matrix when token length $T = 9$. When $\boldsymbol{P}^h$ in Eqn.(5) is a matrix in (a), DTN aggregates its mean and variance by considering all tokens. Hence, DTN becomes IN in this case. When $\boldsymbol{P}^h$ in Eqn.(5) is a matrix in (b) or (c), DTN becomes a local version of IN, which obtains the mean and variance for each token by only looking at 3 or 5 neighboring tokens.

$$\begin{bmatrix} 0 & 1 \\ -1 & 0 \end{bmatrix} \quad \begin{bmatrix} 0 & 1 & 0 & 1 \\ -1 & 0 & -1 & 0 \\ 0 & 1 & 0 & 1 \\ -1 & 0 & -1 & 0 \end{bmatrix} \quad \begin{bmatrix} 0 & 0 & 1 & 1 \\ 0 & 0 & 1 & 1 \\ -1 & -1 & 0 & 0 \\ -1 & -1 & 0 & 0 \end{bmatrix}$$

(a) Index of image patch     (b) Horizontal shift $\delta^x$     (c) Vertical shift $\delta^y$

Figure 5: Illustration of construction of horizontal and vertical in relative positional embedding $R$. Take $T = 4$ as an example; tokens with a length of 4 are encoded from a $2 \times 2$ image patches. (a) is index matrix indicating the relative position of each patch. The entries in the index matrix are defined by $j - i$ where $i$ and $j$ are row and column index, respectively. Based on this definition, (b) and (c) show the relative horizontal and vertical shifts between patch $i$ and patch $j$, respectively.



Figure 6: Visualization of $\boldsymbol{P}^h$ initialized by (a) truncated normal distribution and (b-e) Eqn.(7) before training. We visualize the weights corresponding to 91-th token, i.e. $\boldsymbol{P}^h[91,:].\text{view}(14,14)$. The lighter is the larger. A yellow box highlights the underlying token. (a) shows that truncated normal initialization leads to uniform weights for all heads. On the contrary, initialization in Eqn.(7) can give larger weights to tokens in the neighbourhood with size $2 \times 2$. Moreover, each head gives the largest weight to different neighboring token denoted by the red box.

## A MORE DETAILS ABOUT APPROACH

### A.1 ARCHITECTURAL DETAIL

We integrate DTN into various vision transformers such as ViT, PVT and Swin which is instantiated by stacking multiple transformer encoders. Note that our proposed DTN utilizes the relative positional encoding, implying that the class token can not be directly concatenated with the tokens encoded from image patches. To tackle this issue, we employ DTN in the first $l^{dtn}$ transformer encoders and leave the last $L - l^{dtn}$ transformer unchanged where $L$ denotes the number of transformer encoders. In this way, the class token can then appended to the output tokens of the $l^{dtn}$-th transformer encoder by following d'Ascoli et al. (2021); Wang et al. (2021b). By default, we set $l^{dtn} = 10$ for ViTs and $l^{dtn} = 3$ for PVT which indicates that DTN is used in the first 3 blocks of PVT. Since Swin do not utilize class token, we insert DTN into all the blocks of Swin. Note that $l^{dtn}$ determines how many DTNs are plugged into the transformer. We investigate the influence of the number of DTN used in transformers by the ablation study on $l^{dtn}$ as shown in Sec.B.4.

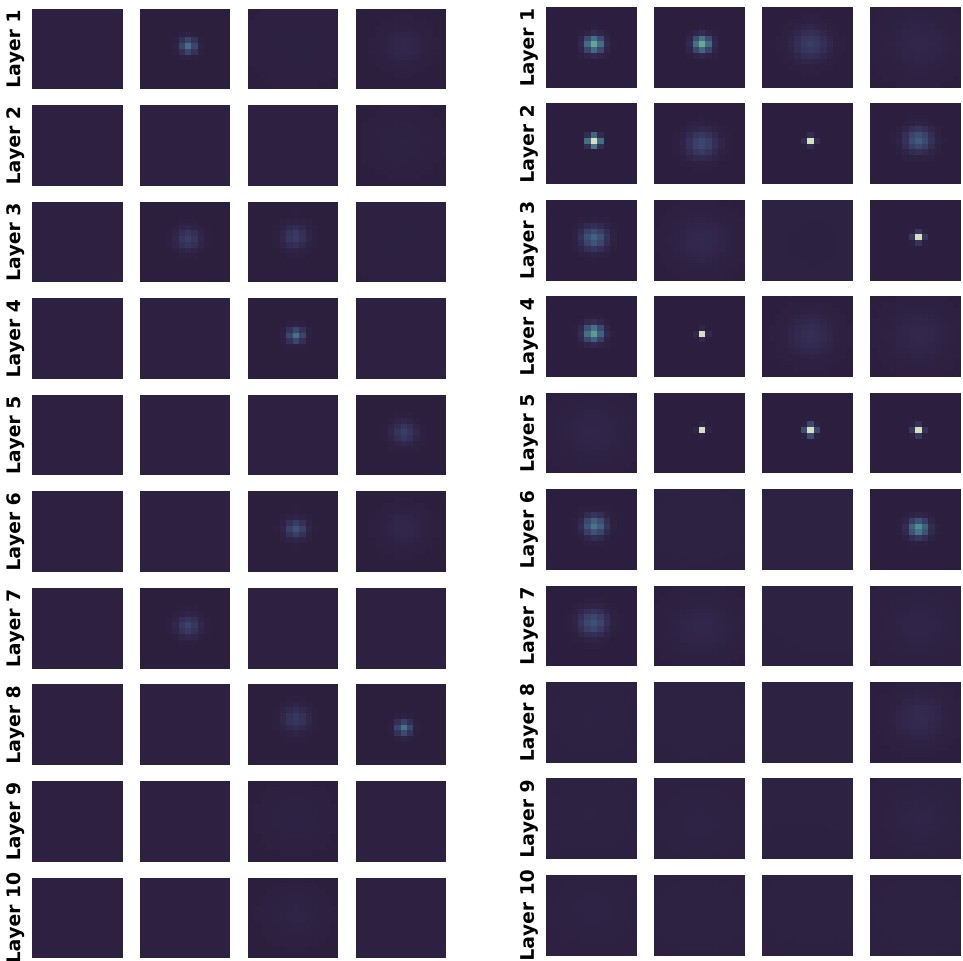

Figure 7: Visualization of $\boldsymbol{P}^h$ initialized by (Left) truncated normal distribution and (Right) Eqn.(7) after training. We visualize the weights corresponding to 91-th token, i.e. $\boldsymbol{P}^h[91, :].\text{view}(14,14)$ for all layers. Lighter is larger.

---

**Algorithm 1** Forward pass of DTN.

---

1: **Input:** mini-batch inputs $\boldsymbol{x} \in \mathbb{R}^{B \times T \times C}$, where the $b$-th sample in the batch is $\boldsymbol{x}^b \in \mathbb{R}^{T \times C}, b \in \{1, 2, ..., B\}$; learnable parameters $\omega^h$ to generate $\lambda^h, h \in [H]$; relative positional embedding $\boldsymbol{R} \in \mathbb{R}^{T \times T \times 3}$ and $\boldsymbol{a}^h \in \mathbb{R}^{3 \times 1}$ in Eqn.(6) in Eqn.(7); scale and shift parameters $\boldsymbol{\gamma}, \boldsymbol{\beta} \in \mathbb{R}^{C \times 1}$.
2: **Initialization:** initialize $\omega^h = 0$ and $\boldsymbol{a}^h = [-1, 2\Delta_1^h, 2\Delta_2^h]^{\mathsf{T}}$ before training.
3: **Hyperparameters:** $\epsilon$.
4: **Output:** the normalized tokens $\{\tilde{\boldsymbol{x}}_n, n = 1, 2, ..., N\}$.
5: calculate: $\lambda^h = \text{sigmoid}(\omega^h)$.
6: calculate: $\boldsymbol{P}^h = \text{softmax}(\boldsymbol{R}\boldsymbol{a}^h)$.
7: **for** $b = 1$ **to** $B$ **do**
8:     calculate: $\mu_t^{ln} = \frac{1}{C} \sum_{c=1}^{C} x_{tc}^b, (\sigma^2)_t^{ln} = \frac{1}{C} \sum_{c=1}^{C} (x_{tc}^b - \mu_t^{ln})^2, t \in [T]$.      ▷ intra-token statistics.
9:     calculate: $(\boldsymbol{\mu}^{it})^h = \boldsymbol{P}^h(\boldsymbol{x}^b)^h, ((\sigma^2)^{it})^h = [\boldsymbol{P}^h((\boldsymbol{x}^b) \odot (\boldsymbol{x}^b)) - (\boldsymbol{P}^h(\boldsymbol{x}^b) \odot \boldsymbol{P}^h(\boldsymbol{x}^b))]$.      ▷ inter-token statistics.
10:     calculate: $\boldsymbol{\mu}^h = \lambda^h(\boldsymbol{\mu}^{ln})^h + (1 - \lambda^h)(\boldsymbol{\mu}^{it})^h, h \in [H]$.      ▷ mean of DTN.
11:     calculate: $(\boldsymbol{\sigma}^2)^h = \lambda^h((\boldsymbol{\sigma}^2)^{ln})^h + (1 - \lambda^h)((\boldsymbol{\sigma}^2)^{it})^h, h \in [H]$.      ▷ variance of DTN.
12:     concatenate statistics in all heads: $\boldsymbol{\mu} = \text{Concate}_{h \in [H]}\{\boldsymbol{\mu}^h\}, \boldsymbol{\sigma}^2 = \text{Concate}_{h \in [H]}\{(\boldsymbol{\sigma}^2)^h\}$.
13:     calculate DTN output: $\tilde{\boldsymbol{x}} = \boldsymbol{\gamma} \odot (\boldsymbol{x} - \boldsymbol{\mu})/\sqrt{\boldsymbol{\sigma}^2 + \epsilon} + \boldsymbol{\beta}$.
14: **end for**

---

# B  MORE EXPERIMENTAL RESULTS

## B.1  IMPLEMENTATION DETAIL

**ImageNet.** We evaluate the performance of our proposed DTN using ViT models with different sizes on ImageNet, which consists of 1.28M training images and 50k validation images. Top-1 and

Table 6: Mean Corruption Error (mCE) of ViT and ResNet models using different normalizers on ImageNet-C, which is designed to evaluate robustness when 'natural corruptions' are presented. The column of 'Norm.' indicates the normalizer. Res-50 is ResNet-50 model. The mCE is obtained by averaging corruption error across all corruption types. Our proposed DTN consistently improves the robustness of ViT-S and ViT-S* by a large margin.

| Norm. | Arch | mCE | Noise | | | Blur | | | | Weather | | | | Digital | | | |
|---|---|---|---|---|---|---|---|---|---|---|---|---|---|---|---|---|---|
| | | | Gauss. | Shot | Impulse | Defocus | Glass | Motion | Zoom | Snow | Frost | Fog | Bright | Contrast | Elastic | Pixel | JPEG |
| BN | Res-50 | 76.7 | 80 | 82 | 83 | 75 | 89 | 78 | 80 | 78 | 75 | 66 | 57 | 71 | 85 | 77 | 77 |
| LN | Vit-S | 42.8 | 40 | 42 | 42 | 50 | 60 | 46 | 57 | 43 | 38 | 39 | 25 | 36 | 44 | 42 | 38 |
| DTN | Vit-S | **40.6** | **39** | **40** | **40** | **49** | **58** | **44** | **54** | **40** | **36** | **34** | **24** | **32** | **43** | **39** | **35** |
| LN | Vit-S* | 41.9 | 40 | 42 | 42 | 51 | 59 | 45 | 56 | 42 | 37 | 36 | 25 | 34 | 44 | 39 | 36 |
| DTN | Vit-S* | **38.0** | **36** | **37** | **37** | **45** | **55** | **41** | **54** | **36** | **35** | **31** | **23** | **30** | **41** | **34** | **33** |

Top-5 accuracies are reported on ImageNet. By default, we set the patch size of ViT models as 16. Moreover, we utilize the common protocols, i.e., number of parameters and Float Points Operations (FLOPs) to obtain model size and computational consumption. We train ViT with our proposed DTN by following the training framework of DeiT (Touvron et al., 2021) where the ViT models are trained with a total batch size of 1024 on all GPUs. We use Adam optimizer with a momentum of 0.9 and weight decay of 0.05. The cosine learning schedule is adopted with the initial learning rate of 0.0005. We use global average pooling for PVT and Swin, where pooling sizes for the first two blocks are set to 4 and 2, respectively.

**ListOps on LRA.** For all experiments on the LRA benchmark, we follow the open repository released by (Tay et al., 2021) and keep all of the settings unchanged. Specifically, all evaluated models have an embedding dimension of 512, with 8 heads and 6 layers. The models are trained for 5K steps on 2K length sequence with batch size 32.

**Definition of mean attention distance in Fig.2.** The mean attention distance is defined by $d = \frac{1}{T}\sum_{i=1}^{T} d_i, d_i = \sum_{j=1}^{T} A_{ij}\delta_{ij}$ where $A_{ij}$ and $\delta_{ij}$ indicate the self-attention weight and Euclidean distance in 2D spatial between token i and token j, respectively. We calculate the mean attention distance for each head by averaging a batch of samples on the ImageNet validation set. When computing the attention weight between token $i$ and other tokens, we deem token $i$ as the attention center. Since the sum over $j$ of $A_{ij}$ is 1, $d_i$ indicates the number of tokens between the attention center token $i$ and other tokens. Therefore, a large mean attention distance implies that self-attention would care more about distant tokens relative to the center token. In this sense, self-attention is thought to model global context. On the contrary, a small mean attention distance implies that self-attention would care more about neighboring tokens relative to the center token. In this case, self-attention can better capture local context.

## B.2 RESULTS ON IMAGENET-C AND IMAGENET-R

DTN can calculate the normalization constants for each token embedding. Thus, DTN is expected to be robust to input perturbations as the mean and variance are specific to each token. To verify this, we compare the robustness to input perturbations of ViT models with LN (baseline) and our DTN. To capture different aspects of robustness, we rely on different, specialized benchmarks ImageNet-C (Hendrycks & Dietterich, 2019) and ImageNet-R (Hendrycks et al., 2021), which contains images in the presence of 'natural corruptions' and 'naturally occurring distribution shifts,' respectively. For the experiment on ImageNet-C, we can see from Table 6 that DTN can surpass the baseline LN by 2.2% mCE metric on ViT-S and 3.9% mCE metric on ViT-S*. Moreover, DTN is also superior to LN in the presence of various natural corruptions such as 'Nosie' and 'Blur' as shown in Table 6. For the experiment on ImageNet-R, DTN obtains 30.7% and 32.7% top-1 accuracy on ViT-S and ViT-S* respectively, which are superior to LN by 1.5% and 3.0% top-1 accuracy. Therefore, our DTN is more robust to natural corruptions and real-world distribution shifts than LN.

## B.3 RESULTS ON SELF-SUPERVISION

A special way of improving a neural model's generalization is supervised contrastive learning [9, 11, 25, 31]. We couple DTN with the supervised contrastive learning on ViT-S* for 100 epoch pre-training, followed by a lineal evaluation. We fine-tune the classification head by 100 epochs

Table 7: **(a)** self-supervised learning with DTN in DINO framework. We pre-train ViT-S$^*$ with DTN for $20, 40, 60, 80$ and $100$ epochs and report top-1 accuracy on linear evaluation. DTN is also effective in learning representation in self-supervision. **(b)** Comparisons of parameter and computation complexity between different. $B, T, C, H$ are # samples, # tokens, embedding dimension of each token, and # heads, respectively.

(a)

| Epochs | 20 | 40 | 60 | 80 | 100 |
|---|---|---|---|---|---|
| ViT-S (LN) | 58.8 | 67.2 | 70.1 | 72.8 | 74.0 |
| ViT-S$^*$ (LN) | 58.5 | 68.3 | 71.3 | 74.0 | 74.7 |
| ViT-S$^*$ (DTN) | **58.9** | **68.8** | **72.0** | **74.4** | **75.2** |

(b)

| Method | Complexity | |
|---|---|---|
| | # parameters | computation |
| LN | $2C$ | $O(BCT)$ |
| IN | $2C$ | $O(BCT)$ |
| ScaleNorm | $C$ | $O(BCT)$ |
| DTN | $2C + 3H$ | $O(BCT^2)$ |

Table 8: Effect of the number of DTN layers on ViT-S$^*$. More DTN leads to better recognition performance.

| $l^{dtn}$ | 0 (LN) | 4 | 6 | 8 | 10 (DTN) |
|---|---|---|---|---|---|
| Top-1 acc | 80.6 | 81.2 | 81.4 | 81.5 | **81.7** |

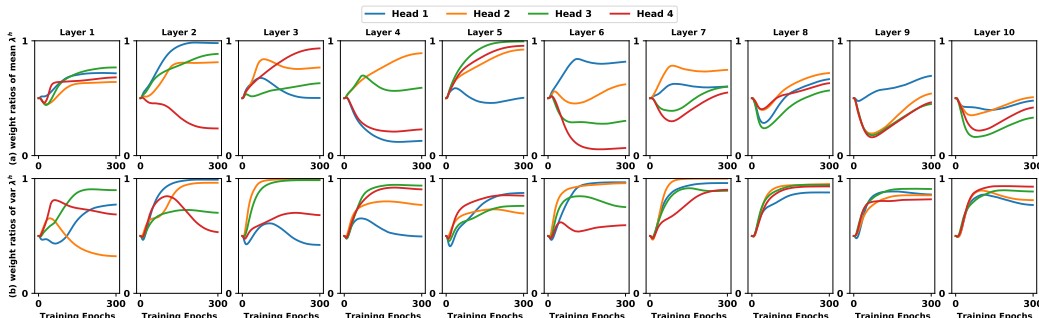

Figure 8: Training dynamics of $\lambda^h$ for mean and variance in different layers of ViT-T$^*$.

for both ViT-S with LN. Please see the Appendices for more implementation details. Compared to the training procedure with LN, we find considerable performance gain as our DTN can encourage transformers to capture both the long-range dependencies and the local context, improving the ImageNet top-1 accuracy of ViT-S$^*$ from 74.7% to 75.2%.

### B.4 MORE ABLATION STUDIES

**Learning dynamics of $\lambda^h$.** In the implementation, we treat the weight ratios $\lambda^h$ of mean and variance in Eqn.(5) differently because mean and variance in normalization play a different role in network's training. For DTN layers of ViT-T$^*$, we plot the learning dynamics of $\lambda^h$ for (a) mean and (b) variance. As implied by Eqn.(5), the smaller $\lambda^h$ is, the more important inter-token statistics would be. As shown in Fig.8, we have three observations. First, the weight ratio of mean and variance have distinct learning dynamics. $\lambda^h$ of the mean for different heads are more diverse than that of variance. Second, different DTN layers have different learning dynamics, which are smoothly converged in training. Third, multiple heads in shallow layers prefer inter-token statics. Whereas larger $\lambda^h$ are typically presented in higher layers. It is consistent with Fig.2 where some heads in shallow layers have a small attention distance while most heads in higher layers have a large attention distance.

**Effect of the number of DTN layers.** As discussed in Sec.A.1, $l^{dtn}$ determines how many DTNs are plugged into the transformer. We investigate the influence of the number of DTN used in transformers by the ablation study on $l^{dtn}$. We set $l^{dtn} = 4, 6, 8, 10$ for VIT-S$^*$ which means LN in the first $4, 6, 8, 10$ transformer encoders are replaced by our DTN while leaving the remaining transformer encoder layers unchanged. The results are obtained by training VIT-S$^*$ with different $l^{dtn}$'s on ImageNet. As shown in Table, more DTN layers bring greater improvement on classification accuracy.

Table 9: Performance of DTN on larger models.

|  | PVT-Large | | | Swin-S | | | PVTv2-B3 | | |
| --- | --- | --- | --- | --- | --- | --- | --- | --- | --- |
|  | Top-1 | Params. | FLOPs | Top-1 | Params. | FLOPs | Top-1 | Params. | FLOPs |
| Baseline | 81.7 | 61.4M | 9.8G | 83.0 | 50M | 8.7G | 83.2 | 45.2M | 6.9G |
| DTN | **82.3** | 61.4M | 10.5G | **83.5** | 50M | 9.4G | **83.7** | 45.2M | 7.4G |

Table 10: Ablation study of DTN on relative positional embedding (RPE) on ImageNet. Ablation (a) denotes that we investigate the effect of RPE in eq.6 of DTN on models without RPE such as ViT and PVT. Ablation (b) indicates that DTN can still improve vision transformers with carefully designed RPE.

| Ablation | Method | Model | Top-1 (%) | Model | Top-1 (%) | Ablation | Method | Model | Top-1 (%) |
| --- | --- | --- | --- | --- | --- | --- | --- | --- | --- |
| (a) | LN + MHSA | ViT-S* | 80.6 | PVT-Tiny | 75.1 | (b) | LN | Swin-S | 83.0 |
|  | LN + MHSA w/ Eqn.(6) | ViT-S* | 80.8 | PVT-Tiny | 75.4 |  | DTN | Swin-S | 83.5 |
|  | DTN w/o RPE + MHSA | ViT-S* | 81.4 | PVT-Tiny | 75.9 |  | LN | LeViT-128S | 76.7 |
|  | DTN + MHSA (ours) | ViT-S* | **81.7** | PVT-Tiny | **76.3** |  | DTN | LeViT-128S | 77.3 |

**Performance of DTN on larger models.** we employ DTN on some large-scale models, including Swin-S, PVT-L, and PVTv2-B3 (Wang et al., 2021a). Note that these models either have large sizes ($> 45M$) or attain outstanding top-1 accuracy ($> 82.5\%$) on ImageNet. As shown in Table 9, our DTN achieves consistent gains on top of these large models. For example, DTN improves the plain Swin-S and PVT-L by 0.5% and 0.6% top-1 accuracy, respectively. For the improved version of the PVT model, e.g., PVTv2-B3(Wang et al., 2021a), we also observe the performance gain ($+0.5\%$ top-1 accuracy).

**How RPE (i.e. $R$ in Eqn.(6) affects the performance of DTN.** We clarify that our DTN is a normalization component that helps a variety of transformer models (ViT, PVT, Swin, T2T-ViT, BigBird, Reformer, etc.) learn better token representation, even some of the models use a relative positional encoding. To separate the effect of RPE in Eqn.(6) from DTN, we conduct two ablation studies on Imagenet as shown in Table 10. (a) We put RPE in Eqn.(6) into the self-attention module of ViT and PVT and use LN to normalize the tokens. Note that neither ViT nor PVT uses RPE in their implementation. Hence, ablation (a) can evaluate how RPE in Eqn.(6) affects the performance of the original model. We also investigate the effect of RPE in Eqn.(6) by removing it from DTN; (b) We plug DTN into vision transformers using carefully-designed RPE such as LeViT(Graham et al., 2021) and SWin (Liu et al., 2021), which demonstrates that DTN can still improve the models with RPE. **Results.** From ablation (a) in Table 10, we can see that a naive RPE used in Eqn.(6) has a marginal improvement when directly put into the MHSA module. However, the performance boosts a lot when such simple RPE is used in our DTN as did in Eqn.(6). We also see that removing RPE in Eqn.(6) has limited influence on the performance of DTN. Moreover, our DTN can still improve those models with carefully-designed RPE as shown by ablation (b) in Table 10, demonstrating the versatility of DTN as a normalization technique.

**Results on downstream task.** We assess the generalization of ourDTN on detection/segmentation track using the COCO2017 dataset ( Lin et al. (2014)). We train our model on the union of 80k training images and 35k validation images and report the performance on the mini-val 5k images. Mask R-CNN and RetinaNet are used as the base detection/segmentation framework. The standard COCO metrics of Average Precision (AP) for bounding box detection (APbb) and instance segmentation (APm) is used to evaluate our methods. We use MMDetection training framework with PVT models as basic backbones and all the hyper-parameters are the same as Wang et al. (2021b). Table 11 shows the detection and segmentation results. The results show that compared with vanilla LN, our DTN block can consistently improve the performance. For example, our DTN with PVT-Tiny is 1.4 AP higher in detection and 1.2 AP higher in segmentation than LN. To sum up, these experiments demonstrate the generalization ability of our DTN in dense tasks that require local context.

**The effect of DTN on the token direction and magnitude.** In our DTN, we have an interesting observation about the learning of direction and magnitude. We perform PCA visualization of tokens before and after normalization. The projected tokens in first two principle components are visualized in Fig.9. Results are obtained on the first layer of trained ViT-S with LN and DTN, respectively. It can be seen that the PCA projected tokens after DTN are closer to each other than LN. It implies that DTN encourages tokens to learn a less diverse direction than LN since tokens normalized by DTN have already presented diverse magnitudes. As we know, learning diverse magnitudes (1 Dimension) could be easier than learning diverse directions (C-1 Dimension). Hence, our DTN can reduce the

Table 11: Object detection performance on COCO val2017 using Mask R-CNN and RetinaNet. DTN can consistently improve both AP and box AP by a large margin.

| Backbone | Norm. | RetinaNet 1x | | | | | | Mask R-CNN 1x | | | | | |
|---|---|---|---|---|---|---|---|---|---|---|---|---|---|
| | | AP | $AP_{50}$ | $AP_{75}$ | $AP_S$ | $AP_M$ | $AP_L$ | $AP^b$ | $AP^b_{50}$ | $AP^b_{75}$ | $AP^m$ | $AP^m_{50}$ | $AP^m_{75}$ |
| PVT-Tiny | LN | 36.7 | 56.9 | 38.9 | 22.6 | 38.8 | 50.0 | 36.7 | 59.2 | 39.3 | 35.1 | 56.7 | 37.3 |
| PVT-Tiny | DTN | **38.0** | 58.7 | 40.4 | 22.7 | 41.2 | 50.9 | **38.1** | 60.7 | 41.0 | **36.3** | 57.9 | 38.7 |
| PVT-Small | LN | 40.4 | 61.3 | 43.0 | 25.0 | 42.9 | 55.7 | 40.4 | 62.9 | 43.8 | 37.8 | 60.1 | 40.3 |
| PVT-Small' | DTN | **41.8** | 62.9 | 44.6 | 26.4 | 45.3 | 56.0 | **41.7** | 64.5 | 44.3 | **39.0** | 61.2 | 41.8 |

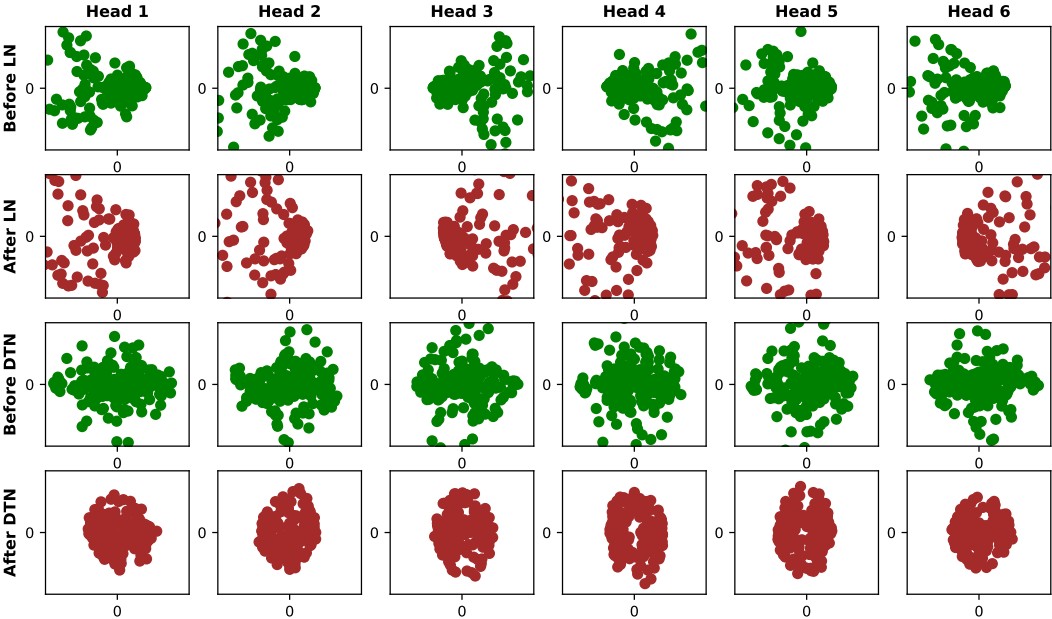

Figure 9: Visualization of token directions using PCA for LN and DTN. The projected tokens in the first two principle component are visualized before and after LN (rows 1-2), and before and after DTN (rows 3-4). We see PCA projected tokens after DTN are closer to each other than LN. It implies that DTN encourages tokens to learn a less diverse direction than LN. Results are obtained on the first layer of trained ViT-S with LN and DTN, respectively.

Table 12: Performance of the proposed DTN evaluated on transformers with local context already induced. DTN can consistently outperform LN with various vision transformers.

| Model | Method | FLOPs | Params | Top-1 (%) |
|---|---|---|---|---|
| Swin-T | LN | 4.5G | 29M | 81.2 |
| | DTN | 5.1G | 29M | **81.9** |
| Swin-S | LN | 8.7G | 50M | 83.0 |
| | DTN | 9.4G | 50M | **83.5** |
| LeViT-128S | LN | 305M | 7.8M | 76.6 |
| | DTN | 320M | 7.8M | **77.3** |
| T2T-ViTt-14 | LN | 6.1G | 21.5M | 81.7 |
| | DTN | 6.4G | 21.5M | **82.4** |

optimization difficulty in learning diverse token representations. In the experiment, we also find that ViT models with DTN can converge much faster than LN. Understanding the role of token magnitude and direction in self-attention modules would be a meaningful future research direction.

**DTN is also effective on those models with local context already induced.** Our DTN as a normalization technique can be seamlessly applied to transformer models with local context already encoded. For example, we replace LN with our DTN on T2T-ViT (Yuan et al., 2021), LeViT (Graham et al., 2021), and Swin. These models introduce positional information, convolutions, or window attention to induce inductive bias. As shown in Table 12, DTN can consistently outperform LN on these models. For example, DTN improves plain Swin-T and Swin-S with LN by 0.7% and 0.5% top-1 accuracy, respectively.

