# OpenReview forum: "Dynamic Token Normalization improves Vision Transformers"
_ICLR.cc/2022/Conference — ICLR 2022 Poster_

### Official Review · Reviewer_g5rX · 2021-10-30

**Correctness:** 4
**Technical Novelty And Significance:** 2
**Empirical Novelty And Significance:** 3
**Recommendation:** 6
**Confidence:** 4

**Main Review:**

**Strengths**
1. The norm component is some important ingredient of ViTs and it is worth further investigation and bringing in our attentions
2. An extensive study of different token norm methods for ViTs
3. The writing is clear and easy to understand
4. The results of DTN is generally good across different benchmarks

**Weaknesses**
1. Motivation is not clear and strong sufficiently:
> + Why is the token magnitude so critical for model performance? The objective of norm itself is to restrict the training data (e.g., tokens in ViTs) to some specific range which has been shown to be helpful for model optimization. For learning capacity, the normalised vectors of high-dimension should be well expressive, and hence there is no high necessity to further explore the magnitude dimension (a single dimension).
>+ In Figure 1, I have these concerns:
>>- If one considers that (a) LN reduces the token magnitude difference for the red box case, then (b) IN and (c) DTN would increase the difference, rather than *preserve*.
>>- From (a) it is seen that different heads actually present different patterns. There is no interpretation and insights that why all heads need to be similar in maintaining the token magnitude, and the current bahaviour of LN is not as good or desired.
>>- Using norm to impose inductive bias (e.g. local context) is some unusual and implicit. How does this compare with existing local window attention e.g. [a], [c], [d], [e].

2. Unclear and inaccurate content
>+ Global vs. Local attention: It is unreasonble that when the model can model global context (attention), it will be hard to capture the local context. Local context is just part of global context. At least this is not accurate statement.
>+ With conventional wisdom, norm is usually not considered as the decisive component for choosing between global context and local context. Instead, it is the scope of tokens in computing the attention scores for given a token, i.e., what tokens are used for pairwise attention learning. So this authors need to be further justified this.
>+ Figure 2: how is the mean attention distance computed? In particular, what is the center of attention?


3. Experiments
>+ What positional embedding is used for baseline such as ViTs, PVTs, and Swin-T? Given that DTN uses the relative positional embedding (RPE) at each block, for a fair comparison, RPE should be applied to baseline as well. This also helps to separate the effect of RPE from the proposed DTN in analysis. Besides, RPE is shown to be beneficial for ViTs in some works [b].
>+ How many iterations run for each experiment? In some cases, the margin of DTN over LN/BN is not big, so the variation of different runs may become more important.
>+ As the authors consider DTN as a component that is able to better learn local context, except comparing different normalization designs, more other alternatives (e.g. [a], [c], [d], [e]) that help with local context learning should be considered in comparison. While some of these works are very new, but the authors should at least include some necessary evaluations on this aspect.

**References**

[a] Stand-Alone Self-Attention in Vision Models. NeurIPS 2019.

[b] LeViT: a Vision Transformer in ConvNet’s Clothing for Faster Inference. arXiv 2021.

[c] Twins: Revisiting the Design of Spatial Attention in Vision Transformers. NeurIPS 2021

[d] Transformer in Transformer. NeurIPS 2021

[e] Tokens-to-Token ViT: Training Vision Transformers from Scratch on ImageNet. ICCV 2021



**Summary Of The Paper:**

This work presents a token normalization method in replacement of Layer Norm (LN) and Instance Norm (IN) for vision transformers (ViTs). The motivation is that the authors find that the common normalization used in most existing ViTs, LN, will reduce the difference in token magnitude, and this may lead to failure of capturing positional context and other inductive bias. Hence, they propose a Dynamic Token Normalization (DTN) component by combining LN and relative positional embedding based transformation. DTN can be plug in varying ViTs e.g., ViT, Swin, PVT. The experiments are done on ImageNet, ImageNet-C, ImageNet-R, and ListOps in supervised learning setting, as well as self-supervised pre-training.

**Summary Of The Review:**

It is an interesting point for research in ViTs. The motivation is not sufficiently clear and solid. Some of the claims are not precise as stated above. In the experiments some important details and comparisons are missing, while a variety of experiments have been included including self-supervised learning.

Overall, in the current form this submission is not strong enough for acceptance. I pretty much vote a rejection rate. However, more information from the authors is needed to make a final recommendation.


******* Post-rebuttal Update *********

The authors have well resolved all of my concerns with additional interpretation and experiments.
Overall, this work is novel in taking the normalization aspect for resolving the limitations of existing ViTs in exploring local context, and showing consistent gain for SOTA ViTs, based on interesting observations and solid design. Therefore, I recommend acceptance.

---

> ### Author Response · Authors · 2021-11-16
> **Responses to Reviewer g5rX**
>
> We thank the reviewer for the careful reading of the manuscript and the constructive remarks. We have provided a detailed general response to the concerns of all the reviewers. Please see details at  [_General Response_](https://openreview.net/forum?id=f9MHpAGUyMn&noteId=al3x7yaB7DcU). We address your concerns in detail as below,
>
> ### __Part 1: Motivation is not clear and strong sufficiently.__
>
> __Q1__: “Why is the token magnitude so critical for model performance? The objective of norm itself is to restrict the training data (e.g., tokens in ViTs) to some specific range which has been shown to be helpful for model optimization. For learning capacity, the normalised vectors of high-dimension should be well expressive, and hence there is no high necessity to further explore the magnitude dimension (a single dimension).”
> __A1__: It is a significant question. We claim that token magnitude matters even though it is just a single dimension for two reasons.
>
> _First_, the core component of the transformer is the self-attention module which is sensitive to the input magnitude. For instance, $\mathrm{softmax}(x_1^T[x_1,x_2,x_3])$ obviously has different distribution nature with $\mathrm{softmax}(10x_1^T[10x_1,x_2,x_3])$ where $x_i$ represents the $i$-th token and we increase the magnitude of the first token by 10 times. Therefore, self-attention weights are heavily affected by input token magnitude. It is noteworthy that the input of self-attention is exactly the output of norm for the commonly-adopted PreNorm architecture. We further find that IN and LN have different effects on the token magnitude, as shown in Figure 1 and Figure 2 in the paper, which motivates us to integrate their advantages.
>
> _Second_, both magnitude and direction are crucial for normalization. For example, previous work such as WeightNorm [1] and BN decouples magnitude and direction on a channel basis in CNN, proven to benefit the optimization of CNNs [2]. However, if we discard the magnitude (remove affine transformation in normalization), all channels would have zero mean and unit variance, then the performance would drop a lot [3].
>
> In our DTN, we also have an interesting observation about the learning of direction and magnitude. We perform PCA visualization of tokens before and after normalization and find that the PCA projected tokens after DTN are closer to each other than LN. It implies that DTN encourages tokens to learn a less diverse direction than LN since tokens normalized by DTN have already presented diverse magnitudes.  As we know, learning diverse magnitudes (1 Dimension) could be easier than learning diverse directions (C-1 Dimension). Hence, our DTN can reduce the optimization difficulty in learning diverse token representations. We will show the visualization in the revision. In the experiment, we also find that ViT models with DTN can converge much faster than LN. Understanding the role of token magnitude and direction in self-attention modules would be a meaningful future research direction.
>
> [1] Weight Normalization: A Simple Reparameterization to Accelerate Training of Deep Neural Networks. Salimans et al. NIPS 2017.
> [2] Optimization Theory for ReLU Neural Networks Trained with Normalization Layers. Dukler et al. ICML 2020.
> [3] Training BatchNorm and Only BatchNorm: On the Expressive Power of Random Features in CNNs. Frankle et al. ICLR 2021.
>
> __Q2__: “If one considers that (a) LN reduces the token magnitude difference for the red box case, then (b) IN and (c) DTN would increase the difference, rather than preserve.”
> __A2__: We agree with the reviewer. For the red box case, DTN increases the token magnitude difference. Here we use a ‘mild’ word, preserve, to emphasize that, unlike LN, our DTN does not reduce the token magnitude difference when tokens before normalization already have different magnitudes. If LN is not used in ViTs, tokens in some heads before normalization would already differ in magnitude, as shown in Figure 1 in the paper. After DTN is applied, the variation in magnitude between tokens still presents. Hence, we use a mild word, i.e. ‘preserve’, to describe this case.
>
> __Q3__: “There is no interpretation and insights why all heads need to be similar in maintaining the token magnitude, and the current behavior of LN is not as good or desired.”
> __A3__: The current behavior of LN is that LN would reduce the token magnitude difference for all heads, which is a natural consequence of the operation of LN, as we explained in Sec. 3.1. Such homogeneous token magnitudes reduce the semantic difference between tokens, preventing almost all heads in MHSA from capturing local context. Therefore, there is interpretation and insights that we can encourage some heads to have the diverse token magnitude to capture local context. Our DTN achieves this by enabling all heads to learn to switch between LN and position-aware intra-token normalization.

---

> > ### Author Response · Authors · 2021-11-16
> > **Responses to Reviewer g5rX (Cont'd)**
> >
> > __Q4__: “Using norm to impose inductive bias (e.g. local context) is some unusual and implicit. How does this compare with existing local window attention e.g. [a], [c], [d], [e].”
> > __A4__: Our DTN as a normalization technique can be seamlessly applied to various transformer models, even those with local context already encoded.  For example, we replace LN with our DTN on T2T-ViT [1], LeViT [2], and Swin. These models introduce positional information, convolutions, or window attention to induce inductive bias. As shown in Table 6, DTN can consistently outperform LN on these models. We will put these evaluations to the revision.
> >
> > [1] LeViT: a Vision Transformer in ConvNet’s Clothing for Faster Inference. arXiv 2021.
> > [2] Tokens-to-Token ViT: Training Vision Transformers from Scratch on ImageNet. ICCV 2021
> >
> > ---
> > __Table 6.__  Performance of DTN on Swin, LeViT, and T2T-ViT.
> >
> > | Model| Norm| FLOPs | Params| top-1 acc (%) |
> > | :----: |:----: |:----: |:----: |:----: |
> > |Swin-T | LN | 4.51G|29M|81.2|
> > |Swin-T | DTN | 5.09G |29M|__81.9__|
> > |Swin-S | LN | 8.7G|50M|83.0|
> > |Swin-S | DTN | 9.4G |50M|__83.5__|
> > |LeViT-128S | LN | 305M|7.8M|76.6|
> > |LeViT-128S | DTN | 320M |7.8M|__77.3__|
> > |T2T-ViTt-14 | LN | 6.1G|21.5M|81.7|
> > |T2T-ViTt-14 | DTN | 6.4G |21.5M|__82.4__|
> >
> > ---
> >
> > ### __Part 2: Unclear and inaccurate content.__
> >
> > __Q5__: “Figure 2: how is the mean attention distance computed? In particular, what is the center of attention?”
> > __A5__:  We’re sorry for the insufficient explanations. Firstly, the mean attention distance is defined by $d = \frac{1}{T}\sum_{i=1}^Td_i, d_i=\sum_{j=1}^{T} A_{ij}\delta_{ij}$ where $A_{ij}$ and $\delta_{ij}$  indicate the self-attention weight and Euclidean distance in 2D spatial between token i and token j, respectively. We calculate the mean attention distance for each head by averaging a batch of samples on the ImageNet validation set. When computing the attention weight between token $i$ and other tokens, we deem token $i$ as the attention center. Since the sum over $j$ of $A_{ij}$ is $1$, $d_i$ indicates the number of tokens between the attention center token $i$ and other tokens. Therefore, a large mean attention distance implies that self-attention would care more about distant tokens relative to the center token. In this sense, self-attention is thought to model global context. On the contrary, a small mean attention distance implies that self-attention would care more about neighboring tokens relative to the center token. In this case, self-attention can better capture local context. We'll give a more detailed explanation about the mean attention distance.
> >
> > __Q6__: “Global vs. Local attention: ... At least this is not an accurate statement.”
> > __A6__:  We’re sorry for the less accurate statement. We will make it correct in revision. In essence, we agree that global context covers local context. However, it is difficult for ViT models with LN to encourage their self-attention heads to capture local context under regular training effectively. It can be seen from Figure 2(a), where most heads have large mean attention distances. As we explained in Q5/A5, a large mean attention distance implies that self-attention cares more about the global context. Previous studies such as ConvBERT[1] and Lite Transformer[2] also find that when explicit inductions for locality are imposed, transformers could be efficient at global and local context learning. In this paper, we induce local context from normalization, which has not been well explored by previous work. We also demonstrate that our proposed normalization scheme can improve various transformer models, even those with local context already encoded.
> >
> > [1] Jiang, et al. ConvBERT. NeurIPS 2020.
> > [2] Wu, et al. Lite Transformer. ICLR. 2019.
> >
> > __Q7__: “With conventional wisdom, norm is usually not considered as the decisive component ... So this authors need to be further justified this.”
> > __A7__:  The normalization technique is commonly used to deal with the training of deep models. However, it is also widely applied to representation learning such as domain generalization [1] and style transfer [2] as norms can operate feature distribution.  In ViTs, we notice that the vanilla self-attention module cannot effectively induce local context. We tackle this issue by exploring the input of self-attention, which is the norm's output for the commonly-adopted PreNorm architecture. We find that the output tokens of LN have homogeneous magnitude, reducing the semantic difference of local context. But IN preserves the magnitude between tokens, thus being effective in inducing local context. These observations give us a clear motivation that the norm could be effective in tackling the inductive bias issue of ViTs. Finally, we are glad to see that our DTN is versatile enough to work well in various transformers.
> >
> > [1] Adversarially Adaptive Normalization. Fan et al. CVPR 2021.
> > [2] Batch-Instance Normalization. Nam et al. NeurIPS 2018.

---

> > > ### Author Response · Authors · 2021-11-16
> > > **Responses to Reviewer g5rX (Cont'd)**
> > >
> > > ### __Part 3: Experiments__
> > >
> > > __Q8__: “What positional embedding is used for baseline such as ViTs, PVTs, and Swin-T? Given that DTN uses the relative positional embedding (RPE) at each block, for a fair comparison, RPE should be applied to baseline as well. This also helps to separate the effect of RPE from the proposed DTN in analysis. Besides, RPE is shown to be beneficial for ViTs in some works [b]”
> > > __A8__:  In our experiment, we replaced LN with our DTN while leaving other components such as positional embedding unchanged for fair comparisons. To separate the effect of RPE in eq.6 from DTN, we have conducted a more detailed ablation study on relative positional embedding. Please see details in Table 2 in _General Response (2)_.
> > >
> > > We clarify that our DTN is a normalization component that helps a variety of transformer models (ViT, PVT, Swin, T2T-ViT, BigBird, Reformer, etc.) learn better token representation, whether these models use a relative positional encoding. Specifically,  from ablation (a) in Table 2, we can see that (1) RPE used in eq.6 has very marginal improvement when directly put into the self-attention module of ViT and PVT. Note that both ViT and PVT do not use relative positional encoding in their implementation; (2) when RPE in eq.6 is removed (fix P at a matrix of all ones), DTN still has consistent performance gains on ViT and PVT. Moreover, from ablation (b) in Table 2, our DTN can still improve those models with carefully-designed RPE such as LeViT and Swin. These additional experiments can demonstrate the superiority of DTN as a normalization method.
> > >
> > >
> > > __Q9__: “How many iterations run for each experiment? In some cases, the margin of DTN over LN/BN is not big, so the variation of different runs may become more important.”
> > > __A9__:  As provided in Sec. B.1 in the Appendix of our paper, we follow the same training framework of DeiT. We train all models for 300 epochs on ImageNet. Our DTN achieves consistent gains on various advanced models, including ViT, PVT, Swin, LeViT, T2T-ViT, BigBird, and Reformer, and on various downstream tasks, including self-supervised pre-training,  robustness in ImageNet-C, ListOps in Long Range Arena, and object detection in COCO.
> > >
> > > __Q10__: “As the authors consider DTN as a component that is able to better learn local context, except comparing different normalization designs, more other alternatives (e.g. [a], [c], [d], [e]) that help with local context learning should be considered in the comparison. While some of these works are very new, but the authors should at least include some necessary evaluations on this aspect.”
> > > __A10__:  According to the code of ICLR review, we’re not required to compare our work with those papers released in Arxiv or published in online proceedings in four months. However, we thank the reviewer for the suggestion. It is meaningful to investigate the effect of DTN on the top of recently-proposed advanced models.  We have evaluated DTN on those models with local context already encoded, such as T2T-ViT, LeViT, and Swin. As shown in Table 6, DTN can consistently outperform LN on these models. Please see details in Q4/A4.

---

> ### Comment · Reviewer_g5rX · 2021-11-21
> **Post-rebuttal comments**
>
> Thanks for the detailed response by the authors, including more explanation, discussion, clarification, and additional experiments on relative positional encoding. All my concerns have been well resolved.
>
> I appreciate that the authors take the normalization perspective for solving the local context learning limitation with ViTs by presenting a versatile norm method for improving a wide rage of ViTs models.
>
> This work should be interesting and inspiring to the community of computer vision now with high attention on the development of ViTs, following the developing path of CNNs in the past decade.
>
> In reflection of the authors' improvements according to the comments, I am happy to lift my rate.

---

### Official Review · Reviewer_j3Et · 2021-11-01

**Correctness:** 3
**Technical Novelty And Significance:** 3
**Empirical Novelty And Significance:** 3
**Recommendation:** 6
**Confidence:** 4

**Main Review:**

+1) Both the motivation and idea are clear. The analysis is reasonable.

+2) The ablation study shows that DTN can improve the performance on some small/middle-scale models.



- Q1: Can authors evaluate the training or inference throughput? Flops sometimes cannot reflect model speed.

- Q2: DTN only is evaluated with small models (e.g. swin-tiny, vit-small, vit-base, etc.). It may be more robust if DTN can improve performance on some larger models (e.g. ViT-L, Swin-B, Swin-L, PVT-L, etc) or down-stream tasks (detection, segmentation, recognition, etc.).

- Q3: I notice that the relative positional embedding is used in DTN. However, vanilla ViTs use the learnable positional embedding. Does the positional embedding keep same in ablation studies?

- Q4: The motivation is to balance LN and IN in token normalization. It is straightforward to integrate output of LN and output of IN with a balance weight. However, P^h = softmax(R*a^h) is closely related to positional offsets. Therefore I am afraid the improvements mainly come from positional information instead of normalization paradigms. In other words, ViTs can introduce a better positional embedding but does not change LN to do the same thing.

**Summary Of The Paper:**

The paper first analyzes the limitation of LN in Transformers and then proposes DTN to capture both long-range dependencies and local positional context. DTN is a unified version that balances LN and IN. Extensive experiments show the effectiveness of the proposed DTN with some small/middle-scale Transformers on the ImageNet.

**Summary Of The Review:**

The paper clearly presents the problem and methods. The experiments can be improved to further support the conclusions. Therefore, I rate to 'marginally above the acceptance threshold' now.

---

> ### Author Response · Authors · 2021-11-16
> **Responses to Reviewer j3Et**
>
> We would like to thank the reviewer for the useful and helpful comments on our manuscript. We have provided a detailed general response to the concerns of all the reviewers. Please see details at  [_General Response_](https://openreview.net/forum?id=f9MHpAGUyMn&noteId=al3x7yaB7DcU). We address the reviewer's concern as below,
>
> __Q1__: “Can authors evaluate the training or inference throughput? Flops sometimes cannot reflect model speed.”
> __A1__: Thanks for the suggestion. We compare the throughput (images/sec, per GPU) of LN and DTN on ViT models, as shown in Table 5. The GPU model is Tesla V100, 32G.  We can see that DTN can improve LN on ViT models with different sizes while introducing a marginal increase of computational cost in terms of throughput and FLOPs. For example, DTN surpasses LN by 0.8% -1.1% top-1 accuracy with about 20% - 30% drop in throughput.
>
> ---
> __Table 5.__  Comparison of  throughput (images/sec) between DTN and LN
>
> | Model| Method| throughput (train) | throughput (inference) | FLOPs |top-1 acc (%)|
> | :----: |:----: |:----: |:----: |:----: |:----: |
> |ViT-T* | LN | 545|1381|1.26G|72.3|
> |ViT-T* | DTN (ours) | 378|974|1.40G|__73.3__|
> |ViT-S* | LN | 223|688|5.77G|80.6|
> |ViT-S* | DTN  (ours) | 167|484|6.08G|__81.7__|
> |ViT-B* | LN | 94|289|17.58G|81.7|
> |ViT-B* | DTN  (ours) | 79|232|18.13G|__82.5__|
>
> ---
>
> __Q2__: “DTN only is evaluated with small models (e.g. swin-tiny, vit-small, vit-base, etc.). It may be more robust if DTN can improve performance on some larger models (e.g. ViT-L, Swin-B, Swin-L, PVT-L, etc) or down-stream tasks (detection, segmentation, recognition, etc.).”
> __A2__: Thanks for the suggestion.  We have experimented with DTN on some larger models, including Swin-S, PVT-L, and PVTv2-b3. Note that these models either have large sizes (>45M) or attain outstanding top-1 accuracy (>82.5%) on ImageNet. As shown in Table 1 (see _General Response (1)_), our DTN achieves consistent gains (>= 0.5% top-1 acc) on top of these large models. Please see detail in _General Response (1)_.
>
> We also have demonstrated that DTN can also improve the performance in COCO detection when it is plugged into the backbone. For example, DTN improves the PVT-Tiny backbone with RetinaNet and Mask R-CNN by +1.3 and +1.4 box AP. Please see detail in General Response (3).
>
> __Q3__: “I notice that the relative positional embedding is used in DTN. However, vanilla ViTs use the learnable positional embedding. Does the positional embedding keep the same in ablation studies?.”
> __A3__: Yes, we keep the positional embedding unchanged in vanilla ViTs.  For fair comparisons, when employing DTN on different transformer models, we replace LN with our DTN while leaving other components unchanged. By this setup, we can reasonably verify the effectiveness of our DTN over LN on various models and tasks.  However, we note that ViTs with DTN rely less on absolute positional encoding. As shown in Q2/A2 to Reviewer 2 (PWvF), we can see that DTN has a lower accuracy drop when the model trained with 224 images is evaluated on 384 ImageNet. Hence, ViTs with DTN could remove the absolute positional encoding. In this way, ViT with DTN may not require interpolation of the positional embeddings when changing the input resolution.
>
> __Q4__: “The motivation is to balance LN and IN in token normalization. It is straightforward to integrate the output of LN and the output of IN with a balance weight. However, P^h = softmax(R*a^h) is closely related to positional offsets. Therefore I am afraid the improvements mainly come from positional information instead of normalization paradigms. In other words, ViTs can introduce a better positional embedding but does not change LN to do the same thing.”
> __A4__: We have conducted a more detailed ablation study on relative positional embedding. Please see details in Table 2 in _General Response (2)_. We clarify that our DTN is a normalization component that helps a variety of transformer models (ViT, PVT, Swin, T2T-ViT, BigBird, Reformer, etc.) learn better token representation, whether these models use a relative positional encoding.
>
> Specifically,  from ablation (a) in Table 2, we can see that (1) RPE used in eq.6 has very marginal improvement when it is directly put into the self-attention module of ViT and PVT. Note that both ViT and PVT do not use relative positional encoding in their implementation; (2) when RPE in eq.6 is removed (fix P at a matrix of all ones), DTN still has consistent performance gains on ViT and PVT. Moreover, from ablation (b) in Table 2, our DTN can still improve those models with carefully-designed RPE such as LeViT and Swin. These additional experiments can demonstrate the superiority of DTN as a normalization method.

---

> > ### Comment · Reviewer_j3Et · 2021-11-17
> > **Question about the Table 5**
> >
> > Thanks for your rebuttal. But I am confused about the numbers in Table 5. I think the inference throughput of ViT-S should be approximately 1.8 times that of ViT-B. But the throughput of ViT-T, ViT-S and ViT-B is almost same.

---

> > > ### Author Response · Authors · 2021-11-17
> > > **Re: Question about the Table 5**
> > >
> > > Thanks for the correction. We're sorry for the mistake. We find that the batch size was set to 1 when testing inference speed on GPU, which may not exploit the advantages of GPU, resulting in an inaccurate measurement. Now, we set the batch size to 16 and report the results averaging over 100 runs.  Please see Table 5 again.

---

### Official Review · Reviewer_PWvF · 2021-11-02

**Correctness:** 3
**Technical Novelty And Significance:** 3
**Empirical Novelty And Significance:** 3
**Recommendation:** 5
**Confidence:** 4

**Main Review:**

Pros:
1. The paper is well written and organized.
2. The performance of popular vision transformer models gain with the proposed DTN, no matter with the ViT model or with the stage-wise model PVT and Swin Transformer;
3. The proposed DTN can also be combined with other sparse attention modules such as BigBird or Reformer to boost the performance;
4. The visualization results in Figure 2 confirm that DTN can boost the model to have a relatively small attention distance in early layers.

Cons:
1. As shown in Table 3, the performance of ViT with BN drops a lot. Can you give some more explanations? As BN has proven to be more effective than LN as shown in many CNN networks, I am wondering why it is less effective than LN in vision transformers. The reason that tokens represent different semantics seems not convincing, since contextual information is also very important in vision tasks.
2. It seems that the initialization of DTN depends on the image size, as it uses position embeddings. As demonstrated in CPVT, such learned or fixed position embedding may not be so effective in testing on variant image sizes, i.e., training with image size 224 and test with image size 384, which is necessary for downstream tasks. As the proposed DTN is also integrated into Swin and PVT, can the authors provide the performance comparison between the origin Swin/PVT and the DTN integrated versions on downstream tasks, such as detection/segmentation?
3. All experiments are conducted upon models with performance less than 82.5 top-1 accuracy on ImageNet, which makes the utilization of DTN on more powerful models unclear. The authors should consider more experimental results of DTN on other models like Swin-B or PVTv2-B3.


Additional:

1. The notion of \mu \ sigma in Figure 4 seems to be put at the wrong place, which makes readers confusing.

2. Just for curiosity, if we initialize the DTN modules with the $lambda$ learned in Figure 4, fix them and train the model from scratch, would that lead to better performance?


**Summary Of The Paper:**

This paper studies properties in layer normalization and instance normalization, and states that the two normalization operations have their own drawbacks in vision transformers: LN suffers from lacking inductive bias while IN may be affected by the different semantics within the tokens. Accordingly, this paper proposes a new normalization method called DTN and considers both inter- and intra- token normalization into vision transformers. According to the experiments in the paper, the proposed DTN normalization boosts the performance of various vision transformer models on classification with ImageNet dataset as well as robustness with IMAGENET-C and IMAGENET-R.

**Summary Of The Review:**

The paper's analysis seems reasonable on some given models. However, it is not clear in the paper whether this finding is still valid for larger models, or whether it can handle inputs with different sizes. I am willing to raise my score if the authors can address my concerns in their response.

---

> ### Author Response · Authors · 2021-11-16
> **Responses to Reviewer PWvF**
>
> We thank the reviewer for the detailed comments and valuable suggestions. We have provided a detailed general response to the concerns of all the reviewers. Please see details at  [_General Response_](https://openreview.net/forum?id=f9MHpAGUyMn&noteId=al3x7yaB7DcU). We address the reviewer's concern as follows,
>
> __Q1__: “As shown in Table 3, the performance of ViT with BN drops ... very important in vision tasks.”
> __A1__: BN obtains its normalization constant by aggregating all tokens of a mini-batch of images, which impairs the performance of ViTs for two reasons.
> _Firstly_, similar to IN, BN also considers all tokens in an image, leading to inaccurate estimates of normalization statistics. _Secondly_, BN further involves tokens from different samples in a mini-batch. Previous work [1] pointed out that batch statistics in BN for NLP data have a large variance throughout training, resulting in large gradient magnitude. Hence, BN would impair the training of transformer models.
>
> Moreover, batch statistics in BN characterize the population distribution of training data while the self-attention module works in a data-independent manner. Therefore, BN would make a self-attention module data-dependent, which impairs the contextual modeling ability of the self-attention module for each sample. We believe that investigating why BN impeds the performance of ViTs could be a meaningful and exciting topic.
>
> [1] PowerNorm: Rethinking Batch Normalization in Transformers. Shen et al. ICML 2020.
>
> __Q2__: “It seems that the initialization of DTN depends on the image size, as it uses position embeddings ... testing on variant image sizes, i.e., training with image size 224 and test with image size 384, can the authors provide the performance comparison between the origin Swin/PVT and the DTN integrated versions on downstream tasks, such as detection/segmentation?.”
> __A2__: Thanks for the suggestion. The fixed relative position embedding in eq.6 is effective in testing on variant image sizes. To see this, we directly evaluate the performance of ViT models with DTN on ImageNet with resolution 384. To this end, DTN only needs to resample relative positional encodings $R$ according to Fig.6 in Appendix, which can be easily implemented.
>
> Further, we also fine-tune DTN on ImageNet with resolution 384. For fine-tuning, we use the same training settings in resolution 224 except that the initial learning rate, weight decay, and total epochs are set to 5e-6, 1e-8, and 100, respectively. The results are shown in Table 4. We can see that our DTN has a lower performance drop when the model trained with 224 images is evaluated on 384 ImageNet, implying that ViTs with DTN rely less on absolute positional embedding. Moreover, DTN still surpasses LN by 1.1% top-1 accuracy after fine-tuning.
>
> For experiments of DTN on downstream tasks, we find that DTN improves the PVT-Tiny backbone with RetinaNet and Mask R-CNN by +1.3 and +1.4 box AP. Please see detail in _General Response (3)_.
>
> ---
> __Table 4.__  Performance of DTN on ImageNet with resolution 384x384.
>
> | Model| Method| top-1 acc (244) | top-1 acc (384) | top-1 acc (384) w/ fine-tune |
> | :----: |:----: |:----: |:----: |:----: |
> |ViT-T* | LN | 72.3|69.2|75.7|
> |ViT-T* | DTN | __73.2__|__72.6__|__76.8__|
>
> ---
>
> __Q3__: “All experiments are conducted upon models with performance less than 82.5 top-1 accuracy on ImageNet, which makes the utilization of DTN on more powerful models unclear. The authors should consider more experimental results of DTN on other models like Swin-B or PVTv2-B3”
> __A3__: Thanks for the suggestion. We have experimented with DTN on some larger models, including Swin-S, PVT-L, and PVTv2-b3. Note that these models either have either a large size (>45M) or attain outstanding top-1 accuracy (>82.5%) on ImageNet. As shown in Table 1 (see General Response (1)), our DTN achieves consistent gains (>= 0.5% top-1 acc) on top of these large models. Please see detail in General Response (1).
>
> __Q4__: “The notion of \mu \ sigma in Figure 4 seems to be put at the wrong place, which makes readers confused.”
> __A4__: We’re sorry for the typo. We’ll correct it.
>
> __Q5__: “If we initialize the DTN modules with the ones learned in Figure 4, fix them and train the model from scratch, would that lead to better performance?”
> __A5__: It is an interesting question.  We train ViT-T* with lambda fixed at the values in Fig.4 with the same training settings. Unfortunately, we find that ViT-T* with fixed ‘optimal’ lambda reaches 72.9% top-1 accuracy, which is slightly worse than the performance of the original training scheme (learn lambda during the whole training). We guess that the ViT models may need different token representations at different training stages. As we can see from Figure 4, $\lambda$ in different heads drastically changes as the training goes, implying that ViT models tend to learn a dynamic combination of inter- and intra-token statistics during training.

---

### Official Review · Reviewer_AKdH · 2021-11-02

**Correctness:** 2
**Technical Novelty And Significance:** 3
**Empirical Novelty And Significance:** 3
**Recommendation:** 5
**Confidence:** 4

**Main Review:**

Strengths:

* The motivation of this work is clear and solid. Normalization plays an essential role in both CNN and Transformer-based models, but it does not attract too much attention for current ViT models. This work analyses some properties of token normalization. In Fig 2 and Fig3 (b), it shows us the importance of token statistics diversity.

* The proposed method is technically sound. Adoptingrelative positional embedding to significantly reduce the extra learnable parameters is quite interesting and effective.

* The paper is clearly written and easy to follow.

Weakness:

* The novelty of dynamic normalization is limited. The technology of dynamic networks has been widely adopted for architecture, activation functions, and normalization, especially for CNN models. The authors should discuss some related works [1].

* The experimental results are not sufficient to support this work. One key motivation of this work is to easily induce inductive bias such as local context, which is more important in some downstream vision tasks such as detection and segmentation. So I think the single result in imagenet classification task is not enough. The author also claims that DTN is easy to be plugged into Swin and PVT, so it would be appreciated to see the experiments on more downstream tasks.

* Minors: in Eq (3), the sigma sum over $\mu_c^{in}$ should be T, not C.

[1] Luo P, Zhanglin P, Wenqi S, et al. Differentiable dynamic normalization for learning deep representation[C]//International Conference on Machine Learning. PMLR, 2019: 4203-4211.

**Summary Of The Paper:**

This paper introduces a dynamic normalization, named Dynamic Token Normalization (DTN), to replace the vanilla layer norm in ViT. It learns to normalize tokens in both intra-token and inter-token manners, enabling Transformers to capture both the global contextual information and the local positional context. Experimental results show that DTN can improve some Vision Transformers in ImageNet Classification and Long ListOps tasks.

**Summary Of The Review:**

Overall, I like the motivation and some technology in this work, but my major concern is the insufficient experiments. Thus I stand in " marginally below the acceptance threshold" now and waiting for the author's response.

---

> ### Author Response · Authors · 2021-11-16
> **Responses to Reviewer AKdH**
>
> We thank the reviewer for the constructive feedback and helpful suggestions. We have provided a detailed general response to the concerns of all the reviewers. Please see details at  [_General Response_](https://openreview.net/forum?id=f9MHpAGUyMn&noteId=al3x7yaB7DcU). We address the reviewer's concern as follows,
>
>
> __Q1__: “The novelty of dynamic normalization is limited. The technology of dynamic networks has been widely adopted for architecture, activation functions, and normalization, especially for CNN models. The authors should discuss some related works.”
> __A1__: Thanks for the suggestion. Our work is related to dynamic architectures such as mixture-of-experts (MoE) [1] and dynamic channel gating networks [2]. Similar to these works, DTN also learns weight ratios to select computation units. In DTN, such a strategy is very effective to combine intra- and inter-token statistics. In addition, DTN is substantially different from DN [3], where dynamic normalization can be constructed as the statistics in BN and LN can be inferred from the statistics of IN. However, it does not hold in the transformer because LN normalizes within each token embedding. Compared to SN [4], we get rid of BN, which is empirically detrimental to ViTs. Moreover, we use a position-aware probability matrix to collect intra-token statistics, making our DTN rich enough to contain various normalization methods.
>
>
> __Q2__: “The experimental results are not sufficient to support this work. One key motivation of this work is to easily induce inductive bias such as local context, which is more important in some downstream vision tasks such as detection and segmentation.”
> __A2__: We have demonstrated that DTN can also improve the performance in COCO detection when it is plugged into the backbone.  Note that object detection is a dense task that requires local context. In the experiment, we find that DTN with PVT-Tiny as backbone improves RetinaNet by +1.3 box AP and improves Mask R-CNN by +1.4 box AP and +1.2 mask AP. Please see detail in General Response (3).
>
> __Q3__: “Minors: in Eq (3), the sigma sum over $\mu_{c}^{in}$ should be T, not C..”
> __A3__: Thanks for the advice. We’ll correct it in the revision.
>
> [1] Outrageously Large Neural Networks: The Sparsely-Gated Mixture-of-Experts Layer. Shazeer et al. ICLR 2017.
> [2] Channel Gating Neural Networks. Hua et al. NeurIPS 2019.
> [3] Differentiable Dynamic Normalization for Learning Deep Representation. Luo et al. ICML 2019.
> [4] Switchable Normalization for Learning-to-Normalize Deep Representation. Luo et al. ICLR 2019.

---

### Author Response · Authors · 2021-11-16
**General Response to Reviewers’ Concerns.**

We are thankful to the reviewers for their detailed reviews and thoughtful suggestions on our manuscript. Notably, we are glad that the reviewers agreed that the motivation and idea are clear, that normalization is of great importance to transformers, and our analyses of normalization in ViT are thoughtful, and that DTN achieves some improvements on various models.

In general, there are three concerns to our manuscript, including:
1. The performance of DTN on larger models.
2. Does the performance gain come from the relative positional encoding in eq.6 instead of our DTN?.
3. The performance of DTN on downstream tasks that require local context.

To address the reviewers’ concerns, we tried our best to perform several additional experiments as follows.

### __General response (1): performance of DTN on some larger models__

We thank the reviewers for this suggestion which helps demonstrate the versatility of the proposed DTN. To this end, we employ DTN on some large-scale models, including Swin-S, PVT-L, and PVTv2-B3 [1]. Note that these models either have large sizes (>45M) or attain outstanding top-1 accuracy (>82.5%) on ImageNet. As shown in Table 1, our DTN achieves consistent gains on top of these large models. For example, DTN improves the plain Swin-S and PVT-L by 0.5% and 0.6% top-1 accuracy, respectively. For the improved version of the PVT model, e.g. PVTv2-B3 [1], we also observe the performance gain (+0.5% top-1 ACC).

---
   __Table 1.__ Performance of DTN on larger models.

| Model | Norm | FLOPs |Params. | Top-1 Acc (%)|
| :----:| :----: | :----: |:----: | :----: |
| Swin-S | LN | 8.7G |50.0M | 83.0|
| Swin-S| DTN | 9.4G |50.0M | __83.5__|
| PVT-L | LN | 9.8G |61.4M | 81.7|
| PVT-L | DTN | 10.5G |61.4M | __82.3__|
| PVTv2-B3 | LN | 6.9G |45.2M | 83.2|
| PVT-v2-B3 | DTN | 7.4G |45.2M | __83.7__|
---

### __General response (2): does the performance gain come from the relative positional encoding (RPE) in eq.6 instead of our DTN?__

We clarify that our DTN is a normalization component that helps a variety of transformer models (ViT, PVT, Swin, T2T-ViT, BigBird, Reformer, etc.) learn better token representation, even some of the models use a relative positional encoding. We note that RPE in existing vision transformers is utilized in the self-attention module. However, our DTN uses a naive and fixed RPE (i.e. R in eq.6) to generate the positional probability matrix P. We further employ P to collect inter-token statistics by aggregating neighboring tokens.

To separate the effect of RPE in eq.6 from DTN, we conduct two ablation studies on Imagenet as shown in Table 2. (a) We put RPE in eq.6 into the self-attention module of ViT and PVT and use LN to normalize the tokens. Note that neither ViT nor PVT uses RPE in their implementation. Hence, ablation (a) can evaluate how RPE in eq.6 affects the performance of the original model. We also investigate the effect of RPE in eq.6 by removing it from DTN; (b) We plug DTN into vision transformers using carefully-designed RPE such as LeViT [2] and SWin, which demonstrates that DTN can still improve the models with RPE.

__Results.__ From ablation (a) in Table 2, we can see that a naive RPE used in eq.6 has a marginal improvement when directly put into the MHSA module. However, the performance boosts a lot when such simple RPE is used in our DTN as did in eq.6. We also see that removing RPE in eq.6 has limited influence on the performance of DTN. Moreover, our DTN can still improve those models with carefully-designed RPE as shown by ablation (b) in Table 2, demonstrating the versatility of DTN as a normalization technique.

---
   __Table 2.__ Ablation study of DTN on relative positional embedding (RPE) on ImageNet. Ablation (a) denotes that we investigate the effect of RPE in eq.6 of DTN on models without RPE such as ViT and PVT. Ablation (b) indicates that DTN can still improve vision transformers with carefully designed RPE.

| Ablation |     Method        | Model  | Top-1 Acc (%)|
| :----: |:---- |:----: |:----: |
| (a) | LN + MHSA | ViT-S* | 80.6|
| (a) | LN + MHSA w/ RPE in eq.6 | ViT-S*  |80.8|
| (a) | DTN w/o RPE in eq.6 + MHSA  | ViT-S*  | 81.4|
| (a) | DTN+ MHSA (ours) | ViT-S*  | __81.7__ |
| __Ablation__ |     __Method__       | __Model__  | __Top-1 Acc (%)__|
| (a) | LN + MHSA | PVT-Tiny | 75.1|
| (a) | LN + MHSA w/ RPE in eq.6 | PVT-Tiny  |75.4|
| (a) | DTN w/o RPE in eq.6 + MHSA  | PVT-Tiny | 75.9|
| (a) | DTN+ MHSA (ours) | PVT-Tiny | __76.3__ |
| __Ablation__ |     __Method__       | __Model__  | __Top-1 Acc (%)__|
| (b) | LN  | Swin-S | 83.0|
| (b) | DTN (ours)| Swin-S  |__83.5__|
| (b) | LN  | LeViT-128S | 76.6|
| (b) | DTN (ours)| LeViT-128S  |__77.3__|

---

---

> ### Author Response · Authors · 2021-11-16
> **General Response to Reviewers’ Concerns.  (Cont'd)**
>
> ### __General response (3): performance of DTN on downstream tasks__
>
> Thanks for the suggestion. We have experimented with DTN in object detection on the COCO benchmark with PVT as the backbone. Specifically, we apply the PVT model with our DTN onto two representative dense prediction methods: RetinaNet and Mask R- CNN. Following the training configuration in PVT, we adopt a 1x training schedule (i.e., 12 epochs) to evaluate DTN on the detection task. The results are reported in Table 3. We can see that DTN is also effective in dense downstream tasks. For example, DTN with PVT-Tiny as backbone improves RetinaNet by +1.3 box AP and improves Mask R-CNN by +1.4 box AP and +1.2 mask AP. With more advanced transformers, such as PVTv2-B3, DTN also achieves +0.7 box AP and +0.6 mask AP gains over LN. These results demonstrate the effectiveness of our DTN on dense downstream tasks.
>
> ---
> __Table 3.__ Performance of DTN in COCO detection with PVT model as the backbone.
>
> | Method | Norm| Backbone |box AP |mask AP |
> | :----: | :----: | :----: |:----: |:----: |
> | RetinaNet | LN| PVT-Tiny |36.7 |-- |
> | RetinaNet | DTN| PVT-Tiny |__38.0__ |-- |
> | Mask R-CNN | LN| PVT-Tiny |36.7 |35.1 |
> | Mask R-CNN | DTN| PVT-Tiny |__38.1__ |__36.3__ |
> | Mask R-CNN | LN| PVTv2-B3|47.0 |42.5 |
> | Mask R-CNN | DTN| PVTv2-B3 |__47.7__ |__43.1__ |
> ---
>
> ​​[1] PVTv2: Improved Baselines with Pyramid Vision Transformer. Wang et al. arXiv 2021.
> ​​[2] LeViT: a Vision Transformer in ConvNet’s Clothing for Faster Inference. Graham el al. arXiv 2021.

---

### Author Response · Authors · 2021-11-18
**Paper Update**

We thank the reviewers for their thoughtful and constructive comments. We have made a number of changes to the text according to the suggestions from reviewers. We use the blue font to highlight the changes we have made. Thanks to the reviewers’ constructive feedback, we believe that our paper has been substantially improved.  We list changes we have made as below:

1. Abstract has been updated to emphasize the versatility and effectiveness of our DTN.
2. Some statements have been made clear in the Introduction (i.e. local and global context).
3. The relation between our DTN and dynamic architectures has been discussed in the Related Work.
4. Some typos in Eqn. (3) and Figure 4 have been corrected.
5. More ablation studies have been conducted in Sec. B.4 of Appendix, including the performance of DTN on larger models (Table 9), on downstream tasks such as object detection in COCO (Table 11), and on transformer models with local context already induced (Table 12).
6. A sufficient ablation study on RPE has been done to separate the effect of RPE from our DTN (Table 10).
7. The token direction before and after LN and DTN are analyzed in Sec. B.4. Our DTN reduces the difficulty in learning diverse token representation.
8. The definition of mean attention distance is given in Sec. B.1 of Appendix.
9. To satisfy the page limit, we put the analyses about the learning dynamics of $\lambda$ to Appendix Sec. B.4.

---

### Decision · Program_Chairs · 2022-01-20

**Decision:**

Accept (Poster)

**Comment:**

A new method for dynamic token normalization in ViTs (both within and across tokens) is introduced in the paper. As noted by the reviewers, the proposed method is technically sound, with a clear and solid motivation. The main raised concerns included the lack of experiments using larger models, unclear reason for the accuracy gains, and lack of experiments on other tasks beyond classification, such as detection and segmentation. The authors’ response was strong, clarifying other questions and providing additional experiments, for example, showing the effectiveness of the method on object detection, and when applied to larger models or architectures that explicitly model local context. Two reviewers recommend borderline rejection, but they did not participate in the discussion nor updated their reviews after the author response. The AC considers that their concerns were adequately addressed by the rebuttal, and agrees with the other two reviewers that the paper passes the acceptance bar of ICLR. The authors should carefully proofread the paper for the final version.